# A Theoretical Framework for Inference Learning

**Nick Alonso**[1]*, **Beren Millidge**[3], **Jeff Krichmar**[1,2], **Emre Neftci**[1,2,4]

[1]Department of Cognitive Science, UC Irvine
[2]Department of Computer Science, UC Irvine
[3]MRC Brain Network Dynamics Unit, University of Oxford
[4]Electrical Engineering and Information Technology, RWTH Aachen, Germany
and Peter Grünberg Institute, Forschungszentrum Jülich, Germany

## Abstract

Backpropagation (BP) is the most successful and widely used algorithm in deep learning. However, the computations required by BP are challenging to reconcile with known neurobiology. This difficulty has stimulated interest in more biologically plausible alternatives to BP. One such algorithm is the inference learning algorithm (IL). IL trains predictive coding models of neural circuits and has achieved equal performance to BP on supervised and auto-associative tasks. In contrast to BP, however, the mathematical foundations of IL are not well-understood. Here, we develop a novel theoretical framework for IL. Our main result is that IL closely approximates an optimization method known as implicit stochastic gradient descent (implicit SGD), which is distinct from the explicit SGD implemented by BP. Our results further show how the standard implementation of IL can be altered to better approximate implicit SGD. Our novel implementation considerably improves the stability of IL across learning rates, which is consistent with our theory, as a key property of implicit SGD is its stability. We provide extensive simulation results that further support our theoretical interpretations and find IL achieves quicker convergence when trained with mini-batch size one while performing competitively with BP for larger mini-batches when combined with Adam.

## 1 Introduction

Backpropagation (BP) [33], the most successful and ubiquitously used learning algorithm in deep learning, has long been criticized as being incompatible with known neurobiology [10, 21]. For example, unlike the highly interconnected, recurrent circuits in the brain, BP performs credit assignment through a separate feedback stream that does not alter feedforward (FF) signals [10, 21]. Additionally, standard implementations of BP are largely incompatible with energy efficient, neuromorphic hardware, due, in part, to BP's non-local gradient computations. These concerns have motivated neuroscientists and engineers to search for alternatives to BP that use more biologically compatible, local learning rules.

One alternative to BP is the inference learning algorithm (IL) [32, 43, 37]. IL is the standard learning algorithm used to train predictive coding (PC) models of biological neural circuits (e.g. [32, 13, 39, 4, 1, 7]). Within neuroscience, PC models have grown in popularity, where some even propose PC may be a canonical biological circuit [15, 14, 4]. IL works by first minimizing an energy function, known as free energy, w.r.t. neuron activities using the recurrent computations involved in predictive coding. At convergence, IL updates weights to further minimize energy. Unlike standard BP, IL uses local learning rules and recurrent circuits instead of a separate feedback stream to perform

---

*Correspondence to nalonso2@uci.edu

36th Conference on Neural Information Processing Systems (NeurIPS 2022).

credit assignment. In addition to evidence the brain does predictive coding (and thus may do IL), recent empirical work has found evidence the brain may learn by first inferring target neural activities through recurrent processing then updating synapses to consolidate those neural activities, similar to IL [38]. IL has also been used to train deep networks for machine learning tasks. IL, for example, achieved comparable classification and recall accuracy to BP trained networks on supervised and auto-associative tasks (e.g, [43, 2, 36, 35], see also below).

Despite IL's wide use in biological models and success on machine learning tasks, a rigorous formal understanding of IL's optimization properties and their differences from BP is lacking. Providing such a characterization could yield several useful contributions: 1) A basis for a novel theory of how optimization and credit assignment may work in the brain, 2) a formal basis for improving, developing, and applying IL to machine learning tasks and neuromorphic hardware.

In this paper, we develop a novel theoretical framework for IL that describes its optimization properties and their similarities and differences from BP. More specifically, 1) we expand beyond current literature by deriving a general form of the IL algorithm, which we call Generalized IL (G-IL). 2) We demonstrate our main result, which is that G-IL closely approximates an optimization method known as *implicit stochastic gradient descent* (implicit SGD), where the approximation is closest in the case of mini-batch size 1. Implicit SGD is distinct from explicit SGD (the standard form of SGD used in machine learning and the sort implemented by BP). 3) We provide theoretical guarantees concerning IL's stability and a connection to Gauss-Newton optimization. 4) We identify a learning rule and variable settings needed in order for IL to equal implicit SGD and use this result to develop a novel implementation of IL called IL-prox. 5) Finally, we present extensive simulation results that support our theoretical findings, and for the first time show certain performance advantages of IL over BP, such as better stability across learning rates and improved convergence speed with biologically realistic online learning tasks. These results collectively suggest that, not only does IL better fit biological constraints, but it is also mathematically justified and has performance advantages over BP in more biologically realistic training scenarios.

## 2   Related Works

**Inference Learning and Backpropagation** Whittington and Bogacz [43] proved that parameter updates performed by the IL algorithm approach those of BP as the global loss approaches zero and optimized activities approach FF activities. This proof also implies, however, that non-zero IL updates essentially never equal those of BP, and suggests IL poorly approximates BP early in training when the loss is large and optimized activities deviate significantly from FF activities. This leaves open the question of whether there is a better description of IL's optimization strategy and the question of how IL is able to minimize the loss in a stable manner early in training. Other work has shown IL is formally related to variational expectation maximization [25, 22], which is a learning algorithm with convergence guarantees [27]. This analysis is a step forward though leaves open the question of how IL relates to BP and the broader framework of gradient-based methods that are the backbone of deep learning. The focus of the current work is to develop insights into these open questions. More recently, several works have altered IL to more closely approximate SGD and BP [26, 37]. These variants significantly change the original IL algorithm, and some of the alterations seem hard to reconcile with neurobiology [26, 37]. Alternatively, instead of altering IL to better fit SGD and BP, we show the standard implementation of IL already closely approximates a working optimization method known as implicit SGD, and we identify the variable settings under which the standard implementation of IL better approximates implicit SGD.

**Similar Algorithms** Algorithms similar to IL have been proposed. For example, the alternating minimization algorithm [9], which was developed independently of the predictive coding and IL framework from neuroscience, updates weights to minimize local prediction errors similar to IL (see equation 4 below). However, [9] do not connect their algorithm to implicit SGD. A more recent variant of this algorithm by Qiao et al. [31] show some formal links to proximal operators, an optimization process related to implicit SGD [30] (see appendix B.1). However, they do not interpret the weight updates as performing the proximal update or implicit SGD as we do here with IL. Proximal operators have also recently been incorporated into learning algorithms for deep networks. Frerix et al. [12] developed a BP, proximal algorithm hybrid. Lau et al. [17] developed an algorithm that combined block coordinate descent with the proximal operator. Other works use proximal operators to reduce noise or perform other tasks, then BP is used to to update at convergence

[23, 45]. All of these algorithms, however, are distinct from IL, as they use BP or other non-local gradient information to compute weight updates or local targets.

**Target Propagation** Finally, IL has some relation to target propagation algorithms, in which local target activities are computed and used to compute local error gradients to update weights. Target propagation [5] and difference target propagation [19] utilize approximate Gauss-Newton updates on activities to create local targets [24, 6], while other algorithms compute targets using gradient updates on activities [18, 29, 3, 16]. These algorithms have important differences from IL, e.g., they use different learning rules than IL, which we discuss further below. In sum, previous works have not developed the same mathematical interpretation of IL that we do here and have not developed the same mathematical descriptions of the differences between IL and BP. Further, none of the works mentioned above produced the same empirical findings we do in our simulations below.

## 3 Background

### 3.1 Notation

| Term | Description |
|------|-------------|
| $W_n$ | Weight Matrix, pre-synaptic layer $n$ |
| $h_n$ | Feedforward Activity layer $n$, $h_n = W_{n-1} f(h_{n-1})$ |
| $\hat{h}_n$ | Optimized/Target Activity layer $n$ |
| $\Delta h_n$ | Optimized Activity Change layer $n$, $\Delta h_n = \hat{h}_n - h_n$ |
| $p_n$ | Local Prediction layer $n$, $p_n = W_{n-1} f(\hat{h}_{n-1})$ |
| $e_n$ | Local Error, layer $n$, either $e_n = \hat{h}_n - h_n$ or $e_n = \hat{h}_n - p_n$ |

Table 1: Notation

Notation describing a multi-layered feed-forward (FF) network (MLP) is summarized in table 1. To simplify notation, we assume that the bias is stored in an extra column on each weight matrix $W_n$ and we assume a 1 is concatenated to the end of the pre-synaptic activity vector ($h_n$ or $\hat{h}_n$ depending). All the following results should hold with and without biases.

### 3.2 Generalized Backpropagation

Consider a set of parameters $\theta^{(b)}$ at training iteration $b$ with global loss function $L$. An explicit SGD iteration subtracts the gradient of the loss from the parameters: $\theta^{(b+1)} = \theta^{(b)} - \alpha \frac{\partial L}{\partial \theta^{(b)}}$, with step size $\alpha$. BP [34] is the common algorithm for implementing SGD in deep networks. Here, to allow for easier comparison to IL, we consider an alternative implementation of SGD in deep networks that involves computing local target activities at hidden and output layers, similar to [18], [3], [29]. Under this regime, each weight matrix $W_n$ is updated using the gradient of local loss $l_{n+1} = \frac{1}{2} \|\hat{h}_{n+1} - h_{n+1}\|^2 = \frac{1}{2} \|e_{n+1}\|^2$. Here $\hat{h}_{n+1}$ is the local target, while $h_{n+1}$ is the FF activity. The target at the output layer is computed $\hat{h}_N = h_N - \frac{\partial L}{\partial h_N}$. At hidden layer $n$, the local target is computed

$$\hat{h}_n = h_n - \frac{\partial l_{n+1}}{\partial h_n} = h_n - \frac{\partial L}{\partial h_n}. \tag{1}$$

It can be shown $h_n - \frac{\partial l_{n+1}}{\partial h_n} = h_n - \frac{\partial L}{\partial h_n}$ by first computing $\hat{h}_{N-1}$:

$$
\begin{aligned}
\hat{h}_{N-1} &= h_{N-1} - \frac{\partial l_N}{\partial h_{N-1}} = h_{N-1} - \frac{\partial l_N}{\partial h_N} \frac{\partial h_N}{\partial h_{N-1}} = h_{N-1} + e_N \frac{\partial h_N}{\partial h_{N-1}} \\
&= h_{N-1} + (h_N - \frac{\partial L}{\partial h_N} - h_N) \frac{\partial h_N}{\partial h_{N-1}} = h_{N-1} - \frac{\partial L}{\partial h_{N-1}}
\end{aligned}
\tag{2}
$$

This same process can be applied recursively to show the remaining $\hat{h}_n$ are also global loss gradient steps over $h_n$. See appendix of [3] for similar proof.

Updates to weights $W_n$ also use the gradient of the local loss such that $\Delta W_n = -\alpha \frac{\partial l_{n+1}}{\partial W_n}$. If we use the result from the above derivation that $-\frac{\partial l_n}{\partial h_n} = -\frac{\partial L}{\partial h_n}$, we can see this update is equivalent to a global loss gradient step over the weights:

$$\Delta W_n = -\alpha \frac{\partial l_{n+1}}{\partial h_{n+1}} \frac{\partial h_{n+1}}{\partial W_n} = -\alpha \frac{\partial L}{\partial h_{n+1}} \frac{\partial h_{n+1}}{\partial W_n} = -\alpha \frac{\partial L}{\partial W_n} = \alpha e_{n+1} f(h_n)^T. \tag{3}$$

This local target formulation of SGD is similar to other target propagation (TP) [5, 19] algorithms, which use essentially the same learning rule but compute targets slightly differently. For example, a well-known variant of TP is difference target propagation (DTP) [19], which computes local targets by approximating a Gauss-Newton update on activities rather than gradient updates [24, 6]: $\hat{h}_n \approx h_n - J_{N,n}^+ e_N$, where $J_{N,n}^+$ is the pseudo-inverse of the Jacobian of the forward network from layer $n$ to $N$ and $e_N = y - h_N$. Recent attempts to improve DTP generally do so by making DTP updates more similar to SGD updates (e.g. [24, 11]). Additionally, variants of BP algorithms, like random feedback alignment [20] and sign symmetric feedback alignment [44], approximate SGD. Since all of these methods approximate the (explicit) SGD update over weights and use similar learning rules, we consolidate them all under the heading of generalized BP (G-BP). The target-based formulation of G-BP is shown in algorithm 1.

### 3.3 Generalized Inference Learning

We now present a general description of the IL algorithm, which we call Generalized Inference Learning (G-IL). A summary of the algorithm is presented in algorithm 2. At each layer $n$ of an MLP, G-IL stores two variables: $\hat{h}_n$ and $p_n$. At initialization these variables are set to feed-forward activity values: $\hat{h}_n = p_n = h_n$. Then $\hat{h}_n$ are altered over time to minimize the energy function $F$:

$$F = L(y, \hat{h}_N) + \sum_{n=1}^{N} \gamma_n \frac{1}{2} \|\hat{h}_n - p_n\|^2 + \sum_{n=1}^{N-1} \gamma_n^{decay} \frac{1}{2} \|\hat{h}_n\|^2, \tag{4}$$

where $L$ is the global loss, $\frac{1}{2}\|\hat{h}_n - p_n\|^2$ are local losses, $\gamma_n^{decay} \frac{1}{2}\|\hat{h}_n\|^2$ is an optional regularization term, and $\gamma$ are positive scalar weighting terms. The prediction $p_n$ is computed as $p_n = W_{n-1} f(\hat{h}_{n-1})$. (Note $p_n$ may also be computed as $p_n = f(W_{n-1}\hat{h}_{n-1})$. Our theoretical results use the first formulation, but we find little difference in performance between the two in practice. See simulations below.) G-IL first minimizes $F$ w.r.t. to activities $\hat{h}$ (inference phase). While optimizing $\hat{h}$, predictions also change since $p_{n+1} = W_n f(\hat{h}_n)$. After $\text{argmin}_{\hat{h}} F$ is estimated, $F$ is again minimized, now w.r.t. the weight, i.e. $\text{argmin}_W F$. Typical IL algorithms use SGD to perform the inference phase with either the LMS rule (below) or Adam optimizers to update weights (e.g. [32, 43]). G-IL, on the other hand, is general w.r.t. the optimization processes that approximate $\text{argmin}_W F$ and $\text{argmin}_{\hat{h}} F$. We consider two weight updates in our analysis and simulations below. First, is a local error gradient update, which is equivalent to the least-mean squares (LMS) rule:

$$\text{argmin}_{W_n} F \approx W_n + \Delta W_n = W_n - \alpha \frac{\partial l_{n+1}}{\partial W_n} = W_n + \alpha e_{n+1} f(\hat{h}_n)^T, \tag{5}$$

where $e_{n+1} = \hat{h}_{n+1} - p_{n+1}$ and local loss $l_{n+1} = \frac{1}{2}\|e_{n+1}\|^2$. The LMS rule does not solve $\text{argmin}_W F$ but only approximates it. The next rule, which is equivalent to the normalized least mean squared rule (NLMS) [28] is

$$\text{argmin}_{W_n} F = W_n + \Delta W_n = W_n + \|f(\hat{h}_n)\|^{-2} e_{n+1} f(\hat{h}_n)^T. \tag{6}$$

The NLMS rule performs a gradient update but with an inverse squared $\ell_2$ norm of the pre-synaptic activities as its step size instead of a hyper-parameter. The NLMS rule is the minimum-norm solution to $argmin_W F$ in the case where we use mini-batch size 1 (see proposition B.1). The LMS rule closely approximates the NLMS update in the sense the two are proportional. Comparing these rules to the update rule for G-BP (equation 3) we see the main difference is the pre-synaptic term used in the rule: BP uses FF activity $h_n$, while IL uses optimized/target activity $\hat{h}_n$. This difference leads to significantly different stability properties between the algorithms, as we note below.

**Algorithm 1:** Generalized BP

**begin**
   // Feedforward Pass
   $h_0 \leftarrow x^{(b)}$
   **for** $n = 0$ **to** $N - 1$ **do**
      |  $h_{n+1} \leftarrow W_n f(h_n)$
   **end**
   $\hat{h}_N \leftarrow y^{(b)}$
   // Compute Local Targets
   **for** $n = 1$ **to** $N$ **do**
      |  $\hat{h}_n \approx h_n - \frac{\partial L}{\partial h_n}$
   **end**
   // Update Weight Matrices
   Eqn. 3
**end**

**Algorithm 2:** Generalized IL

**begin**
   // Feedforward Pass
   $\hat{h}_0 \leftarrow x^{(b)}$
   **for** $n = 0$ **to** $N - 1$ **do**
      |  $p_{n+1}, \hat{h}_{n+1} \leftarrow W_n f(h_n)$
   **end**
   // Compute Local Targets
   $(\hat{h}_1, ... \hat{h}_N) \approx \mathrm{argmin}_{\hat{h}} \, F$
   // Update Weight Matrices
   $(W_0, ... W_{N-1}) \approx \mathrm{argmin}_W \, F$, Eqn. 4,5
**end**

## 4 Theoretical Results

### 4.1 Implicit Gradient Descent

Let the set of MLP weight parameters $\theta^{(b)} = [W_0^{(b)}, ... W_{N-1}^{(b)}]$, where the input data is $x^{(b)}$, the output target is $y^{(b)}$, and $b$ is the current training iteration. The explicit SGD update uses the loss gradient produced by the current parameters:

$$\theta^{(b+1)} = \theta^{(b)} - \alpha \frac{\partial L(\theta^{(b)}, x^{(b)}, y^{(b)})}{\partial \theta^b}, \tag{7}$$

where $\alpha$ is step size and $L$ is the loss measure. This update is explicit because the gradient can be readily computed given $\theta^{(b)}, x^{(b)}, y^{(b)}$. The *implicit* SGD update uses the loss gradient of the parameters at the next training iteration:

$$\theta^{(b+1)} = \theta^{(b)} - \alpha \frac{\partial L(\theta^{(b+1)}, x^{(b)}, y^{(b)})}{\partial \theta^{(b+1)}}. \tag{8}$$

This is an implicit update because $\theta^{(b+1)}$ shows up on both sides of the equation. Unlike explicit SGD, $\theta^{(b+1)}$ cannot be readily computed using available quantities. However, the implicit gradient update is equivalent to the solution of the following optimization problem (see appendix equation 14):

$$\theta^{(b+1)} = \mathrm{argmin}_\theta \left( L(\theta, x^{(b)}, y^{(b)}) + \frac{1}{2\alpha} \|\theta - \theta^{(b)}\|^2 \right). \tag{9}$$

This update is equivalent to the proximal operator/update [30]. This proximal update changes parameters $\theta^{(b)}$ in a way that minimizes the loss $L$ and the magnitude of the update, which helps keep $\theta^{(b+1)}$ in the proximity of $\theta^{(b)}$. (For more details on the proximal update and its relation to implicit SGD see appendix B.1).

### 4.2 Main Results

We now present our main result, which is to show G-IL is equivalent to implicit SGD in certain limits that are approximated well in practice. We focus on the case where single data-points (mini-batch size 1) are presented each training iteration. This scenario is more biologically realistic than mini-batching and is the case where IL best approximates implicit SGD. We discuss the case of mini-batching in appendix B.5. We first define a kind of IL algorithm that minimizes the proximal loss (equation 9) w.r.t. neuron activities and show G-IL approaches this algorithm in certain limits.

**Definition 4.1.** *Proximal Inference Learning (IL-prox). An algorithm identical to G-IL (algorithm 2), except activities are optimized according to* $\mathrm{argmin}_{\hat{h}} \, prox = \mathrm{argmin}_{\hat{h}} (L(\theta^*) + \frac{1}{2\alpha} \|\theta^* - \theta^{(b)}\|^2)$ *and the NLMS rule is used to update weights.*

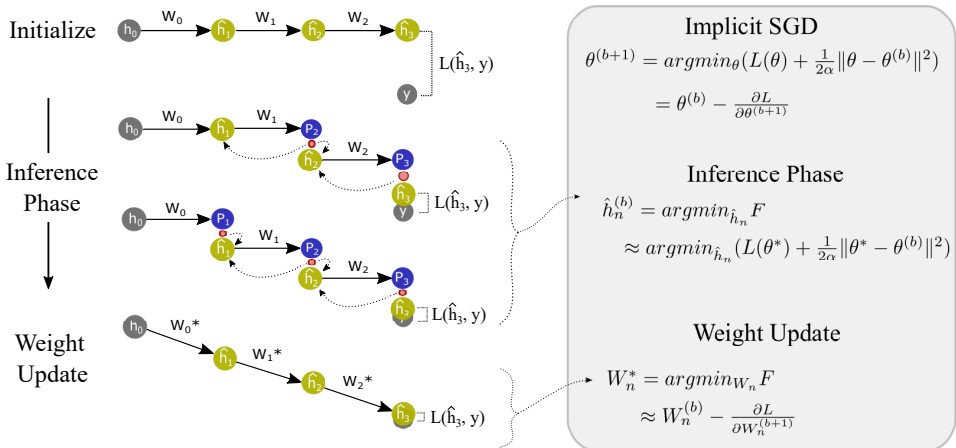

Figure 1: A depiction of IL. Target activities $\hat{h}$ are initialized to FF activities $h$ then updated to minimize energy $F$ during the inference phase. Afterward, weights are updated to further minimize $F$. Weight updates 'align' with $\hat{h}$ activities so $\hat{h}$ become the new FF activities given the same data point. $\hat{h}$ are activities that will both improve the loss $L$ and minimize $\|\Delta\theta\|^2$. This process is equivalent to minimizing the proximal objective and approximates implicit SGD.

Here $b$ is the current training iteration and $\theta^*$ are the parameters updated with the NLMS rule. IL-prox is the same as G-IL except during the inference phase, it minimizes the proximal loss w.r.t. activities $\hat{h}$ instead of the energy $F$. Intuitively, minimizing the proximal loss w.r.t. activities will result in $\hat{h}$ that yields a small loss after weights are updated and small weight update norms. Weight update norm $\frac{1}{2}\|\theta^* - \theta^{(b)}\|^2$ is known during the inference phase because the NLMS learning rule is used and known explicitly:

$$\frac{1}{2}\|\theta^* - \theta^{(b)}\|^2 = \frac{1}{2}\sum_n^N \|\Delta W_n\|^2 = \frac{1}{2}\sum_n^N \|\alpha_n e_{n+1} f(\hat{h}_n)^T\|^2,$$

where $\alpha_n = \|f(\hat{h}_n)\|^{-2}$. Thus, it can be minimized w.r.t. $\hat{h}_n$. As we explain in the appendix (see lemma B.1 and proposition B.1), the loss $L(\theta^*)$ is also explicitly known during the inference phase and can be optimized w.r.t. $\hat{h}_n$.

Now, let $\alpha$ be a 'global' learning rate hyper-parameter, and $\alpha_n$ be layer-wise learning rates used in the actual weight update $\Delta W_n$.

**Theorem 4.1.** *Let $\alpha_n = \|f(\hat{h}_n)\|^{-2}$ and assume mini-batch size 1. In the limit where $\gamma_n^{decay} \to \|e_{n+1}\|^2(1 - \frac{2}{\alpha_n^6})$, $\gamma_N \to \frac{\alpha_{N-1}}{\alpha}$, and $\gamma_n \to \alpha_{n-1}^{-1}$ for all $n < N$, it is the case that $\text{argmin}_{\hat{h}} F = \text{argmin}_{\hat{h}} L(\theta^*) + \frac{1}{2\alpha}\|\Delta\theta\|^2$. Hence, in these limits G-IL is equivalent to IL-prox.*

The proof can be found in supplementary materials theorem B.1. This theorem states that when using the NLMS rule, G-IL is increasingly similar to IL-prox, as the $\gamma$ weighting terms in the free energy (equation 4) approach the scalar values specified in the above theorem. In these limits, $\text{argmin}_{\hat{h}} prox = \text{argmin}_{\hat{h}} F$ and G-IL is equivalent to IL-prox. The intuitive explanation of this theorem goes as follows: minimizing the proximal loss (equation 9) requires minimizing $L$ w.r.t. $\theta^*$ while also minimizing the update norm $\frac{1}{2\alpha}\|\theta^* - \theta^{(b)}\|^2$. The inference phase of IL minimizes $L$ w.r.t. $\theta^*$ by initializing output layer activity $\hat{h}_N$ to FF output $h_N$ then shifting it toward global target $y$. Upon a weight update this will yield a $\theta^*$ that produces a smaller loss $L$. See figure 4.1 for visualization. The magnitude of a weight update $\|e_{n+1}f(\hat{h}_n)^T\|^2$ intuitively depends on the magnitude of local prediction error $e_{n+1}$. F is a sum of the magnitudes of errors (see equation 4), so minimizing F minimizes the magnitudes of errors and consequently weight update norms $\frac{1}{2\alpha}\|\theta^* - \theta^{(b)}\|^2$. Importantly, the $\gamma$ settings under which theorem 4.1 holds can be computed exactly in practice and are of reasonable magnitude that is easily approximated in practice (e.g., they do not approach $\infty$ or 0 and typically approach positive scalars). This is opposed to the limits under which

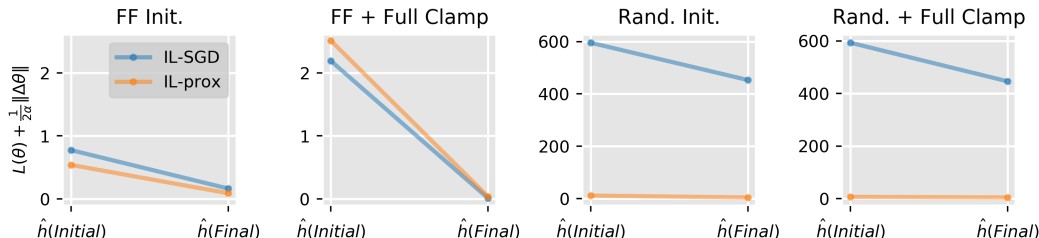

Figure 2: We measure the proximal objective during the inference phase, where F is minimized w.r.t. activities $\hat{h}$, in a two hidden layer MLP with ReLU activations trained on MNIST. We test under four different initializations of $\hat{h}$. First, all $\hat{h}$ are initialized to $h$ (FF Init.). Second, hidden layer $\hat{h}$ are initialized to $h$, while output layer $\hat{h}_N = y$ (FF + Full Clamp). For both of these initializations, the proximal loss is reduced to near zero for both the IL-prox and IL-SGD variants of IL (defined below). We next initialized $\hat{h}$ randomly (Rand Init. and Rand + Full Clamp). Both networks reduce the proximal loss, but nowhere near the global minima, which may help explain why the $\hat{h}$ needs to be initialized to FF activities when $F$ is minimized with SGD. These results support theorem 4.1, which states that minimizing F w.r.t. $\hat{h}$ also minimizes the proximal objective.

IL approximates BP/explicit SGD, which is the limit where $\hat{h}_n \to h_n$ and thus $\Delta W_n \to 0$ [43], a condition that does not generally occur in practice.

**Theorem 4.2.** *Let $\theta^{(b)}$ be a set of MLP parameters at training iteration $b$. Let $\theta_{prox}^{(b+1)} = argmin_\theta L(\theta) + \frac{1}{2\alpha}\|\theta - \theta^{(b)}\|^2$. Let $\theta_{IL-prox}^{(b+1)}$ be the parameters updated by IL-prox (see def. 4.1) and $\theta_{IL}^{(b+1)}$ the parameters updated by G-IL under $\gamma$ values in theorem 4.1. Assume mini-batch size 1. Under this assumption, it is the case $\theta_{prox}^{(b+1)} = \theta_{IL-prox}^{(b+1)} = \theta_{IL}^{(b+1)}$.*

This theorem states that the IL-prox update is equivalent to the proximal update (equation 9), and thus implicit SGD. It follows that, under the $\gamma$ settings in theorem 4.1 the G-IL update is also equivalent to the proximal update/implicit gradient update.

Further theoretical results are developed in the appendix. A novel closed form description of IL targets is developed in C.5. The closed form description shows local targets approximate a regularized Gauss-Newton update on FF activities (see section C). We call this closed form approximation IL-GN. We study the stability of IL-GN when using the LMS learning rule, and compare to an analogous BP network, which uses Gauss-Newton updates (BP-GN). We show that IL-GN weight updates push output layer activities down their loss gradients *for any positive learning rate*, i.e., for any positive learning rate $\hat{h}_N^{(b)} - \hat{h}_N^{(b+1)} = j\frac{\partial L}{\partial \hat{h}_N^{(b)}}$ where $j$ is a positive scalar (theorem D.1). We show BP-GN only has this property for a small range of learning rates (theorem D.2). We develop a partial explanation of why IL is more stable than BP that is based on differences in their learning rule (for details see section E). These results suggest that the differences in the way IL and BP compute and use local targets in their weight updates can explain much of their differences in stability across learning rate. The IL learning rule and target computations are advantageous over that of G-BP in this sense.

## 5   Experiments

In this section, we compare the performance of BP based algorithms to IL algorithms. BP algorithms compute and use explicit gradients to update weights, while IL algorithms compute and use approximate implicit gradients to perform updates. We test BP models that either perform simple gradient descent (**BP-SGD**) or use Adam optimizers (**BP-Adam**). **IL-SGD** is a simple implementation of IL that is similar to implementations of [32, 43]. It uses the LMS rule (local error gradient update) to update weights and optimizes $\hat{h}$ using SGD to near convergence (25 update steps). **IL-Prox** is our novel algorithm that is based on definition 4.1. IL-prox uses the NLMS rule to update weights. Following theorem 4.1, the learning rate is only used to determine how much the output layer activities $\hat{h}_N$ are pushed toward the target $y$. As $\alpha \to \infty$, $\hat{h}_N \to y$ and as $\alpha \to 0$, $\hat{h}_N \to h_N$. IL-Prox

| Reconstruction BCE (mean±std.) | | | | | |
|---|---|---|---|---|---|
| Data | BP-SGD | BP-Adam | IL-SGD | IL-prox | IL-prox fast |
| F-MNIST | .303 | **.282** | .306 | **.286** | **.286** |
| CIFAR-10 | .631 | .622 | .652 | **.611** | **.611** |

Table 3: Best test reconstruction loss after one epoch of training, mini-batch size 1. BCE was averaged across pixels and data-points. Top two scores highlighted in bold. All standard deviations $\leq .002$.

optimizes $\hat{h}$ using SGD to near convergence (25 update steps). **IL-Prox Fast** is the same as the IL-Prox algorithm except it truncates the optimization of $\hat{h}$ to only 12 iterations. **IL-Adam** computes the weight gradients using IL-SGD, then uses Adam optimizers to update weights. **IL-prox Adam** computes the normalized weight gradient using IL-prox and uses Adam optimizers to update weights. All simulations average results over at least 5 seeds of each model type. All models use ReLU activations at hidden layers. Softmax is used at output layer for classification, while sigmoid is used on the autoencoder task.

**Mini-batch Training** Previous work has shown that IL algorithms can achieve the same accuracy as BP on small scale data sets (e.g., MNIST and Fashion-MNIST), in the case where Adam optimizers and medium sized mini-batches (e.g, 64) are used ( [43, 2]. Here we train our BP and IL algorithms on CIFAR-10 with mini-batch size 64. Results are summarized in table 2. Consistent with previous results, IL algorithms, including our novel IL-prox, achieve near equal accuracy as BP-SGD and BP-Adam when using Adam optimizers. However, unlike previous studies, we also test IL-SGD and IL-prox algorithms without

| CIFAR-10 Test Accuracy (mean±std.) | | |
|---|---|---|
| Model | m-batch size=1 | m-batch size=64 |
| BP-SGD | 36.834(±.478) | **57.044** (±.144) |
| IL-SGD | **43.894** (±.371) | 46.924(±.248) |
| IL-prox | 42.060(±.412) | 49.844(±.359) |
| IL-prox Fast | **42.984** (±.530) | 46.094(±.228) |
| BP-Adam | 42.102(±.770) | **56.466** (±.228) |
| IL-Adam | 40.724(±.160) | 54.066(±.190) |
| IL-prox + Adam | 42.802(±.390) | 55.572(±.172) |

Table 2: Best test accuracy on CIFAR-10 supervised task with mini-batch size 1 after 50,000 iterations (1 epoch) and with mini-batch size 64 with 77000 iterations.

Adam, and find they do not perform as well as BP algorithms in this scenario. IL algorithms without Adam tend to converge more quickly but to shallower local minima. Adam, with its momentum and adaptive learning rate, may help prevent IL algorithms from getting caught in shallow local minima (see discussion).

**Online Learning** We find IL algorithms tend to converge significantly quicker and to more similar loss/accuracies as BP in a more biologically realistic, online learning scenario, where a single data-point is presented each training iteration and each data point is seen only once. This is especially apparent with CIFAR-10, shown in tables 2 and 3 and figure 3. IL-SGD and IL-prox algorithms perform similar to and even slightly better than BP-Adam, despite not using momentum or stored, parameter-wise adaptive learning rates like Adam. As far as we know, this is a novel result. Less significant speedups were also seen on MNIST and Fashion-MNIST classification tasks (see figure 6). On autoencoder tasks, IL-prox algorithms, in particular, decreased loss significantly quicker than BP-SGD and even slightly quicker than BP-Adam (figure 3).

| Stability Test: CIFAR-10 Test Accuracy | | | | | | |
|---|---|---|---|---|---|---|
| Model | lr=.01 | lr=.1 | lr=1 | lr=2.5 | lr=10 | lr=100 |
| BP-SGD | 35.48(±1.65) | − | − | − | − | − |
| IL-SGD | 41.66(±1.65) | 36.268(±.271) | − | − | − | − |
| BP-prox | 38.39(±2.91) | 23.978(±1.57) | 12.89(±1.1) | 12.10(±1.61) | 12.21(±1.61) | − |
| IL-prox | 34.97(±.67) | 33.47(±2.56) | 37.38(±1.46) | 37.12(±1.81) | 37.01(±1.01) | 37.64(±.74) |

Table 4: Accuracies after 50,000 training iterations on CIFAR-10, mini-batch size 1. Fully connected networks with layer sizes 3072-3x1024-10.

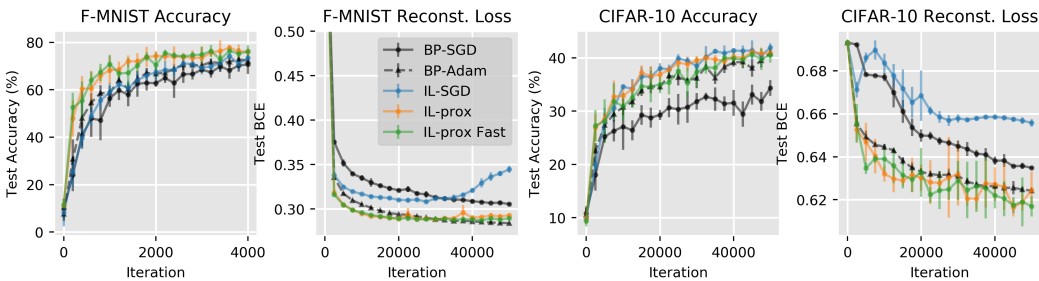

Figure 3: Training runs for mini-batch size 1 (online) scenario, for classification and autoencoder tasks. MLP size 784-2x500-10 and autoencoder dimension 784-256-100-256-784 are trained on F-MNIST for 50000+ iterations. MLP size 3072-3x1024-10 and autoencoder 3072-1024-500-100-500-1024-3072 are trained on CIFAR-10 for 50000+ iterations. The first 4000 training iterations shown for F-MNIST on left. Full training run are shown for other simulations in right three plots.

**Learning Under Data Constraints** We also find that IL algorithms can converge to better accuracy than BP in highly data constrained scenarios. We trained IL algorithms using on only 10, 100, or 500 data points from each category on F-MNIST and CIFAR-10, with mini-batch size 100. IL algorithms achieved better accuracies than BP-SGD in the 10 datapoint scenario on F-MNIST and CIFAR, and better accuracy on the 100 data point scenario on CIFAR, while performances evened out in the 500 data point scenario (see appendix table 8 for details). IL-prox also tended to converge significantly quicker than other algorithms on Cifar-10 (figure 4).

**Stability Test** Explicit SGD is well-known to be highly sensitive to learning rate. Implicit SGD, on the other hand, is highly insensitive to learning rate and is unconditionally stable, i.e., it is able to discount the loss in a stable manner for nearly any positive learning rate [41]. We compare performance of BP-SGD, IL-SGD, and IL-prox on MNIST (table 5) and CIFAR-10 (table 4) classification task across different learning rates. As a control, we train a hybrid of BP and IL-prox (BP-prox). BP-prox, like IL-prox, uses the learning rate only to adjust the target at the output layer and uses the NLMS rule to update weights. Unlike IL-prox, BP-prox computes local targets using the G-BP algorithm rather than the IL algorithm. Results can be found in table 5 and 4. Blank entries are those with accuracies below $12\%$. IL-SGD is more stable than BP-SGD, and IL-prox is more stable than BP-prox. This suggests that the way IL computes and uses targets contributes to stability. Additionally, IL-prox and BP-prox algorithms are more stable than SGD algorithms, showing that using the normalized gradient to update weights and $\alpha$ to adjust the output layer target (rather than scale weight updates) also contributes to stability.

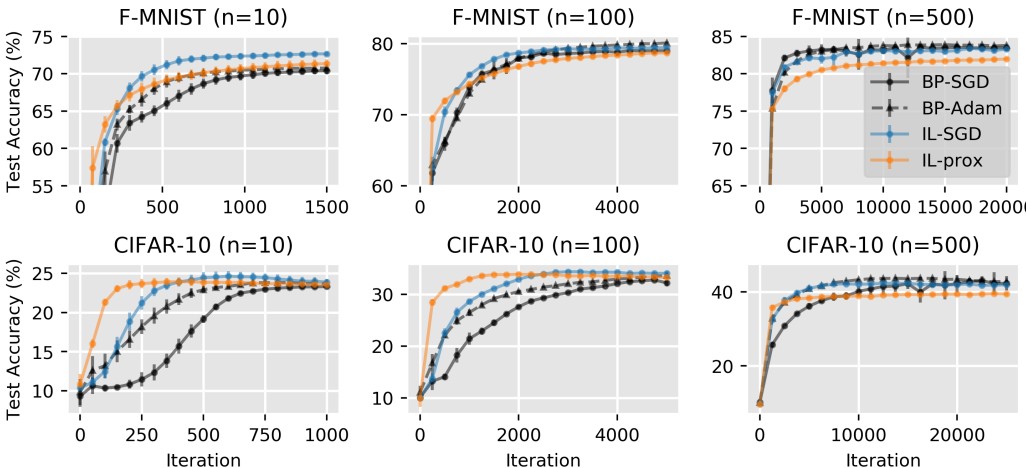

Figure 4: Training runs for a subset of the algorithms on the data constrained scenarios. Each model trained on $n$ data points from each category.

# 6 Limitations

Our theoretical results show that IL algorithms closely approximate implicit SGD in the case where only a single data-point (i.e., mini-batch size 1) is presented each training iteration. This approximation is looser in the case where large mini-batches are used (see appendix B.5). This is no problem from a biological point of view, since the brain does not train with large mini-batches. Additionally, when Adam optimizers are used IL still performs similarly to BP. However, it may still be desirable to further study the optimization properties of IL with mini-batches to improve performance on machine learning tasks where mini-batches are typically used.

# 7 Discussion

In this paper, we found that IL closely approximates implicit SGD, which is distinct from the explicit SGD performed by BP. This theoretical result, along with results that help to characterize how IL and BP behave differently, suggest a hypothesis, which as far was we know is novel: learning in the brain is more similar to implicit gradient descent than explicit gradient descent. This hypothesis would imply the learning characteristics of the brain behave differently from BP and explicit SGD, more generally. More empirical work will be needed to test this theory. However, recent evidence directly supports the idea the brain does something similar to IL [38]. Evidence that neural circuits perform predictive coding provides further, though more indirect, evidence for the idea.

We also found that IL often converged more quickly than BP and achieved similar test losses and accuracies in online and data constrained scenarios. However, in large mini-batch scenarios without data constraints and without the use of Adam optimizers, IL sometimes converged to worse accuracies than BP in classification tasks. With the use of Adam, IL's performance more closely matched that of BP. One possible explanation of this empirical finding may be related to the fact that IL, and implicit SGD more generally, tend to take smaller steps over parameters and a more direct path toward local minima than BP (e.g., see figures 7 and table 9). More direct paths to minima often lead to faster convergence. Smaller update magnitudes may also act as a kind of built-in regularization that aids in difficult, noisy learning scenarios, such as small mini-batch and data constrained scenarios. In cases where regularization is less advantageous (e.g., no data constraints and large mini-batches), IL's small update magnitudes may provide less of an advantage, and its more direct optimization paths may push parameters into shallow, nearby minima. Adam optimizers, with their momentum and adaptive learning rates, may help push parameters out of shallow minima. More work will be needed to fully characterize and explain these behavioral differences, but we believe the work done here provides a good formal basis for doing so.

These results further suggest IL may have advantages over BP in more biologically realistic machine learning tasks, such as lifelong, online learning scenarios, where streams of data are received by the neural network, a single datapoint at a time, and the model must adapt quickly to changes in the environment and minimize the magnitude of weight updates to prevent catastrophic forgetting. This possibility is especially interesting given that, unlike BP, IL's weight updates are both local in space and time and may thus be more compatible with energy efficient neuromorphic hardware, which is well suited for autonomous, embedded systems that run on batteries and receive data streams through sensors, similar to the brain.

# 8 Broader Impacts

This work is largely theoretical. As such, we do not see any potential negative societal impacts.

# 9 Acknowledgements

This work was partially supported by the National Science Foundation (NSF) under grant 1652159.

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
