# Appendices

## A    Preliminaries

The theoretical analyses developed in this appendix focuses on the learning scenario where single data-points are presented to an MLP each training iteration, similar to online learning scenarios. We focus on this scenario in large part because it is more biologically realistic than training with large mini-batches, and a central goal of the paper is to analyze IL in order to better understand how optimization and credit assignment may work in the brain. Other analyses of learning algorithms make similar simplifying assumptions (e.g., see [24]). We discuss the case of mini-batches briefly in section B.5, and our empirical simulations in this paper and previous works in the literature show that IL algorithms are able to perform competitively with BP when trained on mini-batches (e.g., see [43, 2, 35] as well as in the scenario where single data-points are presented. This suggests using IL with large mini-batches is approximated by the case of training with single data points. We leave it to future work to analyze differences in the behavior of IL in the case of small versus large mini-batches in more detail.

Our analyses considers the version of IL that uses local predictions computed $p_{n+1} = W_n f(\hat{h}_n)$ at hidden layers and $p_1 = W_0 \hat{h}_0$ at the input layer. This kind of prediction can be interpreted as a prediction of neuron pre-activations. This is slightly different from predictions of neuron activations: $p_{n+1} = f(W_n \hat{h}_n)$. Although we only analyse the pre-activation version of the prediction, we find the two behave very similarly in practice (e.g., our IL-SGD model uses prediction of activations, while our IL-prox models use predictions of pre-activations and both achieve similar performance on a variety of tasks). Thus, our analyses here should apply approximately to IL models that use the activation prediction equation.

Finally, we will sometimes use two slightly different expressions to describe the loss produced by MLP parameter $\theta$. First, $L(\theta^{(b)}, x^{(b)}, y^{(b)}) = L(h_N^{(b)}, y^{(b)})$ refers to the loss produced by MLP parameters $\theta^{(b)}$ at training iteration $b$, given data point $x^{(b)}$ and prediction target $y^{(b)}$ during the feedforward pass. This is distinct from $L(\hat{h}_N, y^{(b)})$, which refers to the loss between the output layer optimized activities $\hat{h}_N$ and target $y^{(b)}$. Note that both $h_N$ and $\hat{h}_N$ are initialized to $W_{N-1} f(h_{N-1})$. Importantly, assume in MLPs with non-linearities some non-linearity is applied to the activities at the output layer before being inputted into loss. For example, in the case where cross entropy loss is used, one would compute $L(\sigma(h_N^{(b)}), y^{(b)})$, where $\sigma$ may be sigmoid in the case binary cross entropy or softmax in the case a categorical cross entropy loss. To simplify notation, we do not write this $\sigma$ below, but assume it is 'built' into the loss as needed. None of the proofs below are affected by this assumption.

## B    G-IL and Implicit SGD

### B.1    A Brief Introduction to Implicit SGD

Let $\theta^{(b)}$ be a set of parameters, $x^{(b)}$ input data, and global output target is $y^{(b)}$, where $b$ is the current training iteration. The explicit SGD update uses the loss gradient produced by the current parameters:

$$\theta^{(b+1)} = \theta^{(b)} - \alpha \frac{\partial L(\theta^{(b)}, x^{(b)}, y^{(b)})}{\partial \theta^b}, \tag{10}$$

where $\alpha$ is step size and $L$ is the function being minimized. This update is explicit because the gradient can be readily computed given $\theta^{(b)}, x^{(b)}, y^{(b)}$. The *implicit* SGD update uses the loss gradient of the parameters at the next training iteration:

$$\theta^{(b+1)} = \theta^{(b)} - \alpha \frac{\partial L(\theta^{(b+1)}, x^{(b)}, y^{(b)})}{\partial \theta^{(b+1)}}. \tag{11}$$

This is an implicit update because $\theta^{(b+1)}$ shows up on both sides of the equation. Unlike explicit SGD, $\theta^{(b+1)}$ cannot be readily computed using available quantities. One can still perform an implicit gradient update, however, by computing the solution of the following optimization problem:

$$\theta^{(b+1)} = \operatorname{argmin}_\theta (L(\theta, x^{(b)}, y^{(b)}) + \frac{1}{2\alpha} \|\theta - \theta^{(b)}\|^2). \tag{12}$$

This update is known as the proximal algorithm or proximal point method [30] and is equivalent to performing an implicit SGD update in the stochastic setting (see below). This algorithm is a specific application of the more general proximal operator:

$$prox_{\alpha f}(z) = \mathrm{argmin}_\theta (f(\theta) + \frac{1}{2\alpha}\|\theta - z\|^2), \qquad (13)$$

where $f$ is a function with scalar output. We can see the proximal algorithm changes parameters $\theta^{(b)}$ in a way that minimizes the loss $L$ and the magnitude of the update, which helps keep $\theta^{(b+1)}$ in the proximity of $\theta^{(b)}$. We can also see the proximal algorithm only requires using the known quantities $\theta^{(b)}, x^{(b)}, y^{(b)}$.

The equivalence of the proximal algorithm in the stochastic setting to the implicit SGD update (equation 11) can be shown easily using the fact that at the minimum of the proximal objective $\theta = \theta^{(b+1)}$ and the gradient of 12 is zero:

$$0 = \frac{\partial L}{\partial \theta^{(b+1)}} + \frac{1}{\alpha}(\theta^{(b+1)} - \theta^{(b)})$$
$$\theta^{(b+1)} = \theta^{(b)} - \alpha \frac{\partial L}{\partial \theta^{(b+1)}}, \qquad (14)$$

where $L = L(\theta^{(b+1)}, x^{(b)}, y^{(b)})$.

Explicit and implicit SGD have similar convergence guarantees. Analysis of the convergence properties of implicit SGD in linear regression models was done by [41, 42], in generalized linear models by [40]. However, explicit and implicit SGD have importantly distinct stability properties. It is well-known that explicit SGD is highly sensitive to learning rate. Implicit SGD, on the other hand, is *unconditionally stable* in the sense that equation 12 monotonically decreases the loss function $L$ for $0 < \alpha$, as it is implied by equation 12 that $L(\theta^{(b+1)}) + \frac{1}{\alpha 2}\|\theta^{(b+1)} - \theta^{(b)}\|^2 \le L(\theta^{(b)})$. One can also gain an intuition for the unconditional stability of implicit SGD by noting the differences in the way the learning rate $\alpha$ is used in the explicit SGD update (equation 10) versus the proximal algorithm/implicit update (equation 12). In the explicit SGD update $\alpha$ is used to scale the gradient, which clearly implies that as $\alpha \to \infty$ so does the magnitude of the update. However, with the proximal algorithm $\alpha$ only controls the relative weighting of the two terms in equation 12. As $\alpha \to \infty$ we see the regularization term $\frac{1}{2\alpha}\|\theta - \theta^{(b)}\|^2$ is down-weighted to 0. In this case, the updated parameters are simply those that best minimize the loss function $L$, which lead to an update that reduces the loss given $x^{(b)}$. For further details on the stability of explicit and implicit SGD in linear models see [41, 40, 42].

## B.2   G-IL Local Targets as Target FF Activities

In this section, we show that when a single datapoint is presented each training iteration, there are two interesting properties concerning the local targets computed by IL and the NLMS rule used by IL. We express these properties in lemma B.1 and proposition B.1. This lemma and proposition are used in proofs below.

First, we show that IL local targets $\hat{h}_n^{(b)}$, at training iteration $b$, become the FF activities $h_n^{(b+1)}$ at the next training iteration given the same data point when weights are updated to fully minimize local prediction errors. In this sense, IL local targets are target FF activities.

Let $\theta^{(b)}$ be the set of MLP weight matrices $\theta^{(b)} = [W_0^{(b)}, ..., W_{N-1}^{(b)}]$. Let the solution parameters at iteration $b$ be $\theta^* = \mathrm{argmin}_\theta F$, where $F$ is the energy function defined in equation 4. In the IL algorithm, minimizing $F$ w.r.t. weight $W_n$ equates to updating weights in a way that minimizes the local prediction error $e_{n+1} = \frac{1}{2}\|\hat{h}_{n+1} - p_{n+1}\|^2$ (see section 3.3). In the case where the MLP is only presented with a single data-point each iteration, this local learning problem can be solved such that there is zero error. More specifically, let $W_n^*$ be the weight matrix at layer $n$ of $\theta^*$. When training with single data-points, this solution weight matrix has the property $\hat{h}_{n+1}^{(b)} = W_n^* f(\hat{h}_n^{(b)})$ (for an example of one solution with this property see proposition B.1). It follows trivially that $\hat{h}_n$ are equivalent to the FF activities of $\theta^*$ when given the same data-point $x^{(b)}$ as input:

**Lemma B.1.** *Consider a non-linear MLP trained with G-IL at iteration $b$ with mini-batch size 1. Let $h_n^*$ be the FF activities of solution parameters $\theta^*$ at layer $n$ given data-point $x^{(b)}$. It is the case that $h_n^* = \hat{h}_n^{(b)}$.*

*Proof.* At initialization, $\hat{h}_0^{(b)} = x^{(b)}$ and $h_0^* = x^{(b)}$. Thus, $\hat{h}_0^{(b)} = h_0^*$. Next, by definition at the input layer $\hat{h}_1^{(b)} = W_0^* \hat{h}_0^{(b)}$. This implies that $\hat{h}_1^{(b)} = W_0^* \hat{h}_0^{(b)} = W_0^* h_0^* = h_1^*$. This property holds for remaining layers because by definition $\hat{h}_{n+1}^{(b)} = W_n^* f(\hat{h}_n^{(b)})$, which implies $\hat{h}_2^{(b)} = W_1^* f(\hat{h}_1^{(b)}) = W_1^* f(h_1^*) = h_2^*$. This same procedure can then be repeatedly applied to show the same is true of all remaining layers $n$. $\qquad\square$

The solution matrices $W_n^*$ are non-unique in the case where single-data points are used. However, there are unique minimum-norm solution matrices, which can be computed using the NLMS rule (equation 6) as we now show.

**Proposition B.1.** *Consider an non-linear MLP trained with G-IL under the NLMS rule (equation 16) at iteration $b$ mini-batch size 1. For all $n$, the updated matrix $W_n^{(b+1)}$ is the minimum norm solution matrix, i.e. $W_n^{(b+1)} = \operatorname{argmin}_{W_n^{(b+1)}} \frac{1}{2}\|W_n^{(b+1)} - W_n^{(b)}\|^2$, subject to the constraint that $\hat{h}_n^{(b)} = W_n^{b+1} f(\hat{h}_n^{(b)})$.*

*Proof.* Here we follow the proof of [8], which proved a similar result concerning linear regression models trained with a variant of NLMS.

Using the method of Lagrangian multipliers we can rewrite $\operatorname{argmin}_{W_n} \frac{1}{2}\|W_n^{(b+1)} - W_n^{(b)}\|^2$, subject to $\hat{h}_n^{(b)} = W_n^{b+1} f(\hat{h}_n^{(b)})$ as as follows

$$P = \frac{1}{2}\|W_n^{(b+1)} - W_n^{(b)}\|^2 + \lambda e_{n+1}^{(b+1)}, \tag{15}$$

where $\lambda$ is the Lagrangian multiplier and $e_{n+1}^{(b+1)} = \hat{h}_{n+1}^{(b)} - W_n^{(b+1)} f(\hat{h}_n^{(b)})$. First we compute the gradient of P w.r.t. the weights and set to 0,

$$\frac{\partial P}{\partial W_n^{(b+1)}} = W_n^{(b+1)} - W_n^{(b)} + \lambda f(\hat{h}_n^{(b)})^T = 0. \tag{16}$$

The partial w.r.t. the Lagrangian is then

$$\frac{\partial P}{\partial \lambda} = \hat{h}_{n+1}^{(b)} - W_n^{(b+1)} f(\hat{h}_n^{(b)}) = 0. \tag{17}$$

Rearranging 16 we get

$$W_n^{(b+1)} = W_n^{(b)} - \lambda f(\hat{h}_n^{(b)})^T. \tag{18}$$

Substituting 18 into 17 we get

$$
\begin{aligned}
0 &= \hat{h}_{n+1}^{(b)} - (W_n^{(b)} - \lambda f(\hat{h}_n^{(b)})^T) f(\hat{h}_n^{(b)}) \\
\lambda &= -\|f(\hat{h}_n^{(b)})\|^{-2} e_{n+1}^{(b)}
\end{aligned} \tag{19}
$$
.

Finally, substituting the value for $\lambda$ into 18 we get

$$W_n^{(b+1)} = W_n^{(b)} + \|f(\hat{h}_n^{(b)})\|^{-2} e_{n+1}^{(b)} f(\hat{h}_n^{(b)})^T, \tag{20}$$

which is exactly equal to the NLMS rule used in the IL algorithm (equation 6). $\qquad\square$

In sum, when training with single data-points, activities $\hat{h}$ optimized by IL are target FF activities in the sense that they become FF activities after $F$ is minimized completely w.r.t. weights and local errors go to zero. The NLMS rule yields such a solution. One important implication of this is that

the LMS update (equation 5), which is commonly used in practice, is a good approximation of the NLMS solution since the two are proportional.

This lemma and proposition allow us to do something important, which lays the basis for the connection between implicit SGD and G-IL. In particular, this lemma and proposition show us that the network solution $\theta^*$, its FF activities $h_n^*$, and $\Delta\theta$ can be expressed implicitly in terms of $\hat{h}$. For example, lemma B.1 shows that $h_n^* = \hat{h}_n$. This implies that we know what the loss produced by $\theta^*$:

$$L(\theta^*, x^{(b)}, y^{(b)}) = L(\hat{h}_N^{(b)}, y^{(b)}), \tag{21}$$

where $L(\hat{h}_N, y^{(b)})$ is the loss measure, which in the case of MLPs is some measure of the difference between prediction target $y^{(b)}$ and FF output layer value, which for $\theta^*$ is equivalent to $\hat{h}_N$ (as implied by lemma B.1).

Additionally, proposition B.1 shows that each $\Delta W_n$ can be expressed as $\|f(\hat{h}_n^{(b)})\|^{-2} e_{n+1}^{(b)} f(\hat{h}_n^{(b)})^T$, where $e_{n+1}$ again is can be expressed in terms of optimized activities: $e_{n+1} = \hat{h}_{n+1}^{(b)} - W_n^{(b)} f(\hat{h}_n^{(b)})$. This means

$$\frac{1}{2}\|\theta^* - \theta^{(b)}\|^2 = \frac{1}{2}\sum_n^N \|\Delta W_n\|^2 = \frac{1}{2}\sum_n^N \|\|f(\hat{h}_n^{(b)})\|^{-2}(\hat{h}_{n+1}^{(b)} - W_n^{(b)} f(\hat{h}_n^{(b)}))f(\hat{h}_n)^T\|^2. \tag{22}$$

This allows for the possibility to minimize the proximal quantity w.r.t. $\hat{h}_n$. That is, it allows for an IL algorithm, where during the inference phase the algorithm approximates $\operatorname{argmin}_{\hat{h}}(L(\theta^*) + \frac{1}{2\alpha}\|\theta^* - \theta^{(b)}\|^2)$, where $L(\theta^*)$ and $\|\theta^* - \theta^{(b)}\|^2$ are defined implicitly in terms of $\hat{h}$ using equations 21 and 22. At each step in the inference phase $\hat{h}$ are updated to minimize the proximal loss, which results in a change in $\theta^*$, since $\theta^*$ is defined in terms of the $\hat{h}$. After a minimum is reached weights are updated with the NLMS rule such that $\hat{h}$ values become the FF values given the same data point. The inference phase, in other words, finds the FF values of a $\theta^*$ that best minimizes the proximal loss, then uses those FF values to update the actual parameters $\theta^{(b)}$ such that $\theta^{(b+1)} = \theta^*$ (see figure 4.1 for visualization).

Let's define such an algorithm as proximal inference learning (IL-prox), since it, like G-IL, computes $\hat{h}_n$ by minimizing an objective w.r.t. activities, then updates weights afterward:

**Definition B.1.** *Proximal Inference Learning (IL-prox). An algorithm identical to G-IL (algorithm 2), except activities are optimized to approximate $\operatorname{argmin}_{\hat{h}_n}(L(\theta^*) + \frac{1}{2\alpha}\|\theta^* - \theta^{(b)}\|^2)$ and weights are updated using the NLMS rule.*

Again, the $\theta^*$ here is defined implicitly in terms of the $\hat{h}$ values using equations 21 and 22. In the next, section we show that under certain variable settings it is the case that $\operatorname{argmin}_{\hat{h}} F = \operatorname{argmin}_{\hat{h}}(L(\theta^*) + \frac{1}{2\alpha}\|\theta^* - \theta^{(b)}\|^2)$.

### B.3 Generalized Inference Learning Approximates Proximal Inference Learning

Let $\alpha$ be a 'global' learning rate hyper-parameter for $\theta$, and $\alpha_n$ be the layer-wise learning rate used in the weight update $\Delta W_n$. Finally, let $\theta^*$ be defined as the parameters after the NLMS update is applied to each weight matrix, as in the last section.

**Theorem B.1.** *Let $\alpha_n = \|f(\hat{h}_n)\|^{-2}$. In the limit where $\gamma_n^{decay} \to \|e_{n+1}\|^2(1 - \frac{2}{\alpha_n^6})$, $\gamma_N \to \frac{\alpha_{N-1}}{\alpha}$, and $\gamma_n \to \alpha_{n-1}^{-1}$ for all $n < N$, it is the case that $\operatorname{argmin}_{\hat{h}} F = \operatorname{argmin}_{\hat{h}} L(\theta^*) + \frac{1}{2\alpha}\|\theta^* - \theta^{(b)}\|^2$. Hence, in these limits, G-IL is equivalent to IL-prox.*

*Proof.* To prove this statement, we show that $\frac{\partial F}{\partial \hat{h}_n} = 0 \iff \frac{\partial Prox}{\partial \hat{h}_n} = 0$, which implies $\operatorname{argmin}_{\hat{h}} F = \operatorname{argmin}_{\hat{h}} L(\theta^*) + \frac{1}{2\alpha}\|\theta^* - \theta^{(b)}\|^2$. We first compute the gradients of the IL energy function $F$ w.r.t. activities $\hat{h}$, then the gradient of the proximal loss, which is defined in terms of equation 21 and 22.

**Gradient of F** In IL networks, activities are updated with local gradients of $F$, as follows:

At the output layer,

$$\frac{\partial F(\hat{h}_N)}{\partial \hat{h}_N} = \frac{\partial L(y, \hat{h}_N)}{\partial \hat{h}_N} + \gamma_N e_N. \tag{23}$$

and at hidden layers:

$$\frac{\partial F(\hat{h}_n)}{\partial \hat{h}_n} = -\gamma_{n+1} f'(\hat{h}_n) W_n^T e_{n+1} + \gamma_n e_n + \gamma_n^{decay} f'(\hat{h}_n) \hat{h}_n, \tag{24}$$

where $f'(\hat{h}_n) = \frac{\partial f}{\partial \hat{h}_n}$. We see the gradient of the loss $L$ term is only taken w.r.t. the local output layer activity $\hat{h}_N$, and the gradient of prediction error at layers $n$ and $n+1$ are taken w.r.t. to local layer $n$. We assume whatever process is used to minimize $F$ similarly only uses local information such that its minima are described by the above gradients when they equal zero.

**Gradient of Prox w.r.t. Output Layer** We first compute gradients for the output layer. As with G-IL, we assume only local gradients of the proximal loss are used to update the activities. The gradients of the proximal loss w.r.t. output layer activity $\hat{h}_N$ is

$$Prox(\hat{h}_N) = \underbrace{L(\hat{h}_N, y)}_{prox_1} + \underbrace{\frac{1}{2\alpha} \|\Delta W_{N-1}\|^2}_{prox_2}, \tag{25}$$

since $\hat{h}_N$ is local to $L$ and since $W_{N-1}$ are the only weights local to $\hat{h}_N$.

The gradient $\frac{\partial Prox(\hat{h}_N)}{\partial \hat{h}_N}$, can be expressed in terms of $prox_1$ and $prox_2$ as follows: $\frac{\partial Prox(\hat{h}_N)}{\partial \hat{h}_N} = \frac{\partial prox_1(\hat{h}_N)}{\partial \hat{h}_N} + \frac{\partial prox_2(\hat{h}_N)}{\partial \hat{h}_N}$. Clearly, $\frac{\partial prox_1(\hat{h}_N)}{\partial \hat{h}_N} = \frac{\partial L(\hat{h}_N, y)}{\partial \hat{h}_N}$. The second term $\frac{\partial prox_2(\hat{h}_N)}{\partial \hat{h}_N}$ can be computed using the chain rule. First,

$$prox_2(\hat{h}_N) = \frac{1}{2\alpha} \|\Delta W_{N-1}\|^2 = \frac{1}{2\alpha} \|\alpha_{N-1} e_N f(\hat{h}_{N-1})^T\|^2. \tag{26}$$

where $p_N = W_{N-1} f(\hat{h}_{N-1})$, f is an element-wise non-linearity, and $e_N = \hat{h}_N - p_N$. Then using the chain rule we have

$$\frac{\partial prox_2}{\partial \hat{h}_N} = \frac{\partial prox_2}{\partial \Delta W_{N-1}} \frac{\partial \Delta W_{N-1}}{\partial e_N} \frac{\partial e_N}{\partial \hat{h}_N}, \tag{27}$$

where $\frac{\partial prox_2}{\partial \Delta W_{N-1}} = \frac{\alpha_{N-1}}{\alpha} e_n f(\hat{h}_{N-1})^T$ and $\frac{\partial \Delta W_{N-1}}{\partial e_N} \frac{\partial e_N}{\partial \hat{h}_N} = \alpha_{N-1} f(\hat{h}_{N-1})$. Together we get

$$\begin{aligned}
\frac{\partial Prox(\hat{h}_N)}{\partial \hat{h}_N} &= \frac{\partial L(\hat{h}_N, y)}{\partial \hat{h}_N} + \frac{\alpha_{N-1}^2}{\alpha} \|f(\hat{h}_{N-1})\|^2 e_N \\
&= \frac{\partial L(\hat{h}_N, y)}{\partial \hat{h}_N} + \frac{\alpha_{N-1}}{\alpha} e_N.
\end{aligned} \tag{28}$$

Note that here $\alpha_{N-1} = \|f(\hat{h}_{N-1})\|^{-2}$ as noted in the theorem. In the limit where $\gamma_N \to \frac{\alpha_{N-1}}{\alpha}$, this gradient comes to $\frac{\partial L(\hat{h}_N, y)}{\partial \hat{h}_N} + \gamma_N e_N$, which is equivalent to the gradient $\frac{\partial F}{\partial \hat{h}_N}$ (see equation 23).

**Gradient w.r.t. Hidden Layers** Next, we compute $\frac{\partial Prox(\hat{h}_n)}{\partial \hat{h}_n}$ for hidden layers. The components of $Prox$ local to hidden layer $\hat{h}_n$ are

$$Prox(\hat{h}_n) = \underbrace{\frac{1}{2\alpha} \|\Delta W_n\|^2}_{prox_1} + \underbrace{\frac{1}{2\alpha} \|\Delta W_{n-1}\|^2}_{prox_2}. \tag{29}$$

whose gradient can be expressed in terms of $prox_1$ and $prox_2$: $\frac{\partial Prox(\hat{h}_n)}{\partial \hat{h}_n} = \frac{\partial prox_1(\hat{h}_n)}{\partial \hat{h}_n} + \frac{\partial prox_2(\hat{h}_n)}{\partial \hat{h}_n}$. As above, the gradient $\frac{\partial prox_2(\hat{h}_n)}{\partial \hat{h}_n}$ is computed using the chain rule.

$$prox_2 = \frac{1}{2\alpha} \|\Delta W_{n-1}\|^2 = \frac{1}{2\alpha} \|\alpha_{n-1} e_n f(\hat{h}_{n-1})^T\|^2, \tag{30}$$

where $p_n = W_{n-1} f(\hat{h}_{n-1})$ and $e_n = \hat{h}_n - p_n$. Using the chain rule

$$\frac{\partial prox_2(\hat{h}_n)}{\partial \hat{h}_n} = \frac{\partial prox_2}{\partial \Delta W_{n-1}} \frac{\partial \Delta W_{n-1}}{\partial e_n} \frac{\partial e_n}{\partial \hat{h}_n}, \tag{31}$$

where $\frac{\partial prox_2}{\partial \Delta W_{n-1}} = \frac{\alpha_{n-1}}{\alpha} e_n f(\hat{h}_{n-1})^T$ and $\frac{\partial \Delta W_{n-1}}{\partial e_n} \frac{\partial e_n}{\partial \hat{h}_n} = \alpha_{n-1} f(\hat{h}_{n-1})$. Multiplying together we get

$$\frac{\partial prox_2(\hat{h}_n)}{\partial \hat{h}_n} = \frac{\alpha_{n-1}^2}{\alpha} \| f(\hat{h}_{n-1}) \|^2 e_n. \tag{32}$$

Now we derive $\frac{prox_1(\hat{h}_n)}{\hat{h}_n}$:

$$prox_1 = \frac{1}{2\alpha} \| \Delta W_n \|^2 = \frac{1}{2\alpha} \| \alpha_{n-1} e_{n+1} f(\hat{h}_n)^T \|^2, \tag{33}$$

where $p_{n+1} = W_n f(\hat{h}_n)$ and $e_{n+1} = \hat{h}_{n+1} - p_{n+1}$. Using the chain rule

$$\begin{aligned}
\frac{\partial prox_1}{\partial \hat{h}_n} &= \frac{\partial prox_1}{\partial \Delta W_n} \frac{\partial \Delta W_n}{\partial e_{n+1}} \frac{\partial e_{n+1}}{\partial p_{n+1}} \frac{\partial p_n}{\partial \hat{h}_n} \\
&+ \frac{\partial prox_1}{\partial \Delta W_n} \frac{\partial \Delta W_n}{\partial f(\hat{h}_n)^T} \frac{\partial f(\hat{h}_n)^T}{\partial \hat{h}_n} \\
&+ \frac{\partial prox_1}{\partial \Delta W_n} \frac{\partial \Delta W_n}{\partial \| f(\hat{h}_n) \|^{-2}} \frac{\partial \| f(\hat{h}_n) \|^{-2}}{\partial \hat{h}_n}.
\end{aligned} \tag{34}$$

Unlike the previous gradients, we now need propagate the gradient through the learning rate, which is what is done by the term $\frac{\partial prox_1}{\partial \Delta W_n} \frac{\partial \Delta W_n}{\partial \| f(\hat{h}_n) \|^{-2}} \frac{\partial \| f(\hat{h}_n) \|^{-2}}{\partial \hat{h}_n}$ above.

First, $\frac{\partial prox_1}{\partial \Delta W_n} \frac{\partial \Delta W_n}{\partial e_{n+1}} \frac{\partial e_{n+1}}{\partial p_{n+1}} \frac{\partial p_n}{\partial \hat{h}_n} = -\frac{\alpha_n^2}{\alpha} \| f(\hat{h}_n) \|^2 f'(\hat{h}_n) W_n^T e_{n+1}$. Next, $\frac{\partial prox_1}{\partial \Delta W_n} \frac{\partial \Delta W_n}{\partial f(\hat{h}_n)^T} \frac{\partial f(\hat{h}_n)^T}{\partial \hat{h}_n} = \frac{\alpha_n^2}{\alpha} \| e_{n+1} \|^2 f'(\hat{h}_n) \hat{h}_n$. Finally, $\frac{\partial prox_1}{\partial \Delta W_n} \frac{\partial \Delta W_n}{\partial \| f(\hat{h}_n) \|^{-2}} \frac{\partial \| f(\hat{h}_n) \|^{-2}}{\partial \hat{h}_n} = \frac{\alpha_n^2}{\alpha} (\frac{-2 \| e_n \|^2}{\alpha_n^6}) f'(\hat{h}_n) \hat{h}_n$, which results in

$$\begin{aligned}
\frac{\partial prox_1}{\partial \hat{h}_n} &= -\frac{\alpha_n^2}{\alpha} \| f(\hat{h}_n) \|^2 f'(\hat{h}_n) W_n^T e_{n+1} + \frac{\alpha_n^2}{\alpha} \| e_{n+1} \|^2 f'(\hat{h}_n) \hat{h}_n + \frac{\alpha_n^2}{\alpha} (\frac{-2 \| e_{n+1} \|^2}{\alpha_n^6}) f'(\hat{h}_n) \hat{h}_n \\
&= \frac{\alpha_n^2}{\alpha} (-\| f(\hat{h}_n) \|^2 f'(\hat{h}_n) W_n^T e_{n+1} + \| e_{n+1} \|^2 f'(\hat{h}_n) \hat{h}_n + \frac{-2 \| e_{n+1} \|^2}{\alpha_n^6} f'(\hat{h}_n) \hat{h}_n) \\
&= \frac{\alpha_n^2}{\alpha} (-\alpha_n^{-1} f'(\hat{h}_n) W_n^T e_{n+1} + \| e_{n+1} \|^2 (1 + \frac{-2}{\alpha_n^6}) f'(\hat{h}_n) \hat{h}_n)
\end{aligned} \tag{35}$$

Now we substitute our $\frac{\partial prox_1}{\partial \hat{h}_n}$ and $\frac{\partial prox_2}{\partial \hat{h}_n}$ terms back into the gradient of $Prox(\hat{h}_n)$:

$$\frac{\partial Prox(\hat{h}_n)}{\partial \hat{h}_n} = \frac{\alpha_n^2}{\alpha} (-\alpha_n^{-1} f'(\hat{h}_n) W_n^T e_{n+1} + \alpha_{n-1}^{-1} e_n + \| e_{n+1} \|^2 (1 + \frac{-2}{\alpha_n^6}) f'(\hat{h}_n) \hat{h}_n) \tag{36}$$

Finally, we set this gradient equal to 0 (i.e. at a minimum of Prox) and in the limit where $\gamma_n^{decay} \to \| e_{n+1} \|^2 (1 + \frac{-2}{\alpha_n^6})$ and $\gamma_n \to \alpha_{n-1}^{-1}$ for all $n < N$. To simplify notation, we just label this limit as lim

$$\begin{aligned}
0 &= \lim \frac{\alpha_n^2}{\alpha} (-\alpha_n^{-1} f'(\hat{h}_n) W_n^T e_{n+1} + \alpha_{n-1}^{-1} e_n + \| e_{n+1} \|^2 (1 + \frac{-2}{\alpha_n^6}) f'(\hat{h}_n) \hat{h}_n) \\
&= -\gamma_{n+1} f'(\hat{h}_n) W_n^T e_{n+1} + \gamma_n e_n + \gamma_n^{decay} f'(\hat{h}_n) \hat{h}_n.
\end{aligned} \tag{37}$$

When set to zero, $\frac{\alpha_n^2}{\alpha}$ cancels. The result is exactly equal to $\frac{\partial F}{\partial \hat{h}_n}$ (equation 24). It follows that, in these limits, for any hidden layer $n$ it is the case that $\frac{\partial F}{\partial \hat{h}_n} = 0 \iff \frac{\partial Prox}{\partial \hat{h}_n} = 0$. Since the same result holds at the output layer, it follows that $\operatorname{argmin}_{\hat{h}} F = \operatorname{argmin}_{\hat{h}} L(\theta^*) + \frac{1}{2\alpha} \| \Delta \theta \|^2$. Hence, under these $\gamma$ settings G-IL is equivalent to IL-prox. $\qquad \square$

## B.4 G-IL Approximates Implicit SGD

Here we present the theorem showing the IL-prox, and G-IL with the $\gamma$ setting noted in the above theorem, are equivalent to the proximal algorithm and thus implicit SGD. An intuitive way to think about the relation between the typical proximal algorithm and IL-prox is that IL-prox does what the typical proximal algorithm would do, but in reverse order. The standard proximal algorithm $argmin_\theta L(\theta) + \frac{1}{2\alpha}\|\theta - \theta^{(b)}\|^2$ minimizes the proximal loss w.r.t. parameters, then the new parameters can be used to compute new FF values given the data-point $x^{(b)}$. IL prox, on the other hand, first computes the new FF values of the parameters that best minimize the proximal loss given $x^{(b)}$ during the inference phase. Then it updates weights so they become the parameters that best minimize the proximal loss. Since both optimization problems are unconstrained (and thus optimize over the same space of possible parameter values), they yield the same parameter updates in the end.

**Theorem B.2.** *Let $\theta^{(b)}$ be a set of MLP parameters at training iteration $b$. Let $\theta_{prox}^{(b+1)} = argmin_\theta L(\theta) + \frac{1}{2\alpha}\|\theta - \theta^{(b)}\|^2$. Let $\theta_{IL-prox}^{(b+1)}$ be the parameters updated by IL-prox (see 4.1) and $\theta_{IL}^{(b+1)}$ the parameters updated by G-IL under parameter setting in theorem 4.1. Assume mini-batch size 1. Under these assumptions, it is the case $\theta_{prox}^{(b+1)} = \theta_{IL-prox}^{(b+1)} = \theta_{IL}^{(b+1)}$.*

*Proof.* The weight update procedure performed by IL-prox can be described as follows:

$$\theta_{IL-prox}^{(b+1)} = argmin_\theta F(argmin_{\hat{h}} L(\theta^*) + \|\theta^* - \theta^{(b)}\|^2), \tag{38}$$

where $argmin_\theta F$ is computed using the NLMS rule and $\theta^*$ are the parameters updated with the NLMS rule (equation 16). Lemma B.1 and proposition B.1 imply that the $\hat{h}$ produced by $argmin_{\hat{h}} L(\theta^*) + \frac{1}{2\alpha}\|\theta^* - \theta^{(b)}\|^2)$ are the FF activities of $\theta^*$. The fact that $argmin_\theta F$ is computed using the NLMS rule implies that the resulting parameters are the $\theta^*$ of the optimized activities. $\hat{h}$ are unbounded, real-valued vectors, which implies that $argmin_{\hat{h}} L(\theta^*) + \|\theta^* - \theta^{(b)}\|^2$ is an unbounded optimization problem and also implies that the possible $\theta^*$ values are also unbounded (because they can vary over a set of FF values that have unbounded possible values). This implies that $argmin_{\hat{h}} L(\theta^*) + \|\theta^* - \theta^{(b)}\|^2$ outputs the FF values of the parameters $\theta^*$ that best minimize the proximal loss. Then $argmin_\theta F$ uses those FF activities to update $\theta^{(b)}$ such that it become equal to the $\theta^*$ that best minimize the proximal loss. This implies $\theta_{IL-prox}^{(b+1)} = argmin_\theta L(\theta) + \|\theta - \theta^{(b)}\|^2$ and thus $\theta_{prox}^{(b+1)} = \theta_{IL-prox}^{(b+1)}$.

Further, this conclusion implies that under the $\gamma$ parameter settings and learning rates noted in theorem 4.1, the parameter update produced by G-IL, $\theta_{IL}^{(b+1)}$ also equals $\theta_{prox}^{(b+1)}$, since $\theta_{prox}^{(b+1)} = \theta_{IL-prox}^{(b+1)} = \theta_{IL}^{(b+1)}$, given the proof above and theorem 4.1. $\qquad\square$

## B.5 A Note about Mini-batches

Our analysis above focuses on the more biologically realistic scenario where a single data-point is presented each training iteration. In this case, the NLMS rule provides the minimum-norm solution to $argmin_\theta F$ and the LMS update approximates this solution in the sense it is proportional to the NLMS update. However, in the case where mini-batches of size $> 1$ are used, the NLMS rule is not a solution to $argmin_\theta F$. The gradient update of $F$ w.r.t. $W_n$ is

$$W_n^{b+1} = W_n^{(b)} - \alpha_n \frac{\partial F}{\partial W_n} = W_n^{(b)} + \alpha_n(H_{n+1}^{T(b)} - W_n^{(b)}f(H_n^{T(b)}))f(H_n^{(b)}), \tag{39}$$

where $f$ is an element-wise non-linearity and $H_n$ is the mini-batch matrix of neurons activities. Each row of $H_n$ is a $\hat{h}_n$ for one data-point in the mini-batch.

The solution to the local error minimization problem is found by setting the gradient equal to zero and solving for $W_n$:

$$\begin{aligned} 0 &= (H_{n+1}^{T(b)} - W_n f(H_n^{T(b)}))f(H_n^{(b)}) \\ W_n &= H_{n+1}^{T(b)} f(H_n^{(b)})(f(H_n^{T(b)})f(H_n^{(b)}))^{-1}, \end{aligned} \tag{40}$$

This update is generally not equal or proportional to the gradient update. Gradient updates are thus poorer approximations of the solution(s) to local least mean squared problem in the case of mini-batches size $> 1$ than they are in the case with mini-batches size equal to one. This may provide some insight into why the IL models we tested do not show performance advantages over BP-SGD when mini-batches are used as opposed to when single datapoints are used to update weights. Future research could explore alternative learning rules that better approximate the solution above in the mini-batch case.

## C  A Closed Form Description of G-IL Local Targets

In this section, we derive a closed form description of $argmin_{\hat{h}}F$ for linear MLPs. In addition to simplifying notation, we focus on linear networks because we aim to use this closed form description to analyze the stability properties of linear networks trained with IL in the next section.

### C.1  A Brief Intro to Gauss-Newton Optimization and Ridge Regression

Let $\theta$ be a vector of parameters, $y$ a vector of prediction targets, $X$ a data matrix. The linear least squares problem is defined as

$$\operatorname{argmin}_\theta \|y - X\theta\|^2. \tag{41}$$

Its closed form solution is $(XX^T)^{-1}X^Ty$ which equals $X^+y$ when $X$ has linearly independent columns. $X^+$ is the left pseudo-inverse of $X$. For matrix $M$ the left pseudo-inverse has the property $I = M^+M$. When the regularization term $\lambda\|\theta\|^2$ is added to 41 we get ridge regression whose closed form solution is $(XX^T + \lambda I)^{-1}X^Ty$, where $I$ is the identity matrix and $\lambda$ is a scalar.

The Gauss-Newton method is an iterative method used to solve non-linear least squares problems, which has some similarities to the solutions to linear least squares problem just mentioned. Let $e^{(b)}$ be the residuals of the model at $b$, i.e. $e^{(b)} = y^{(b)} - f(x^{(b)}; \theta^{(b)})$, where $f$ is the non-linear model function. The Gauss-Newton (GN) update for $\theta^{(b)}$ is

$$\begin{aligned} \theta^{(b+1)} &= \theta^{(b)} + (JJ^T)^{-1}J^Te^{(b)} \\ &= \theta^{(b)} + J^+e^{(b)}, \end{aligned} \tag{42}$$

where $J$ is the Jacobian of $f$ w.r.t. $\theta^{(b)}$ and $J^+$ is the left pseudo-inverse of $J$. Note the similarity to the closed form solution to the linear least squares. The parameters are iteratively updated until convergence or a near convergent state is reached.

It is also common to add a regularization term to the Gauss-Newton update as follows

$$\theta^{(b+1)} = \theta^{(b)} + (JJ^T + \lambda I)^{-1}J^Te^{(b)}, \tag{43}$$

where $I$ is the identity matrix and $\lambda$ is some scalar. Notice the similarity to the ridge regression solution. The regularization term helps prevent the change to $\theta$ from growing too large. Note that as $\lambda \to \infty$ the update approaches gradient descent: $\theta^{(b+1)} = \theta^{(b)} + J^Te^{(b)}$.

### C.2  A Closed form Description of G-IL Local Targets

Consider a linear MLP trained with G-IL using a squared error global loss. Local targets at hidden layers of such networks are the result of the optimization process that attempts to find

$$\hat{h}_n^{(b)} = \operatorname{argmin}_{\hat{h}_n}(\frac{1}{2}\|\hat{h}_{n+1}^{(b)} - W_n\hat{h}_n\|^2 + \frac{\lambda}{2}\|\hat{h}_n - p_n^{(b)}\|^2), \tag{44}$$

where $\lambda = \frac{\gamma_n}{\gamma_{n+1}}$ which represents the relative weighting of the two terms (see equation 4). Here we ignore the optional decay term in equation 4.

The $\hat{h}_n^{(b)}$ can be expressed in closed form by noting that at the minimum of the expression above $h_n = \hat{h}_n^{(b)}$, and the gradient of the expression equals 0. One can thus compute the gradient, set it equal to zero, and solve for $\hat{h}_n^{(b)}$:

$$
\begin{aligned}
0 &= -W_n^T(\hat{h}_{n+1}^{(b)} - W_n\hat{h}_n^{(b)}) + \lambda\hat{h}_n^{(b)} - \lambda p_n^{(b)} \\
0 &= -W_n^T\hat{h}_{n+1}^{(b)} + (W_n^T W_n + \lambda I)\hat{h}_n^{(b)} - \lambda p_n^{(b)} \\
\hat{h}_n^{(b)} &= (W_n^T W_n + \lambda I)^{-1} W_n^T \hat{h}_{n+1}^{(b)} + \lambda(W_n^T W_n + \lambda I)^{-1} p_n^{(b)}
\end{aligned}
\tag{45}
$$

Notice the term on the left $(W_n^T W_n + \lambda I)^{-1} W_n^T \hat{h}_{n+1}^{(b)}$ is identical to the closed form solution for ridge regression (see previous section), which is a regularized version of a simple target propagation $W_n^+ \hat{h}_{n+1}^{(b)}$. The term on the right $(W_n^T W_n + \lambda I)^{-1} p_n \approx \epsilon p_n$, where $\epsilon$ is a scalar, since $(W_n^T W_n + \lambda I)^{-1}$ is positive definite and approaches a scalar multiple of $I$ as $\lambda \to \infty$. Thus, at the minimum $\hat{h}_n^{(b)}$ closely approximates a weighted average between $p_n$ and the solution to the regularized squared error at layer $n + 1$.

## C.3   Relation to Gauss-Newton Updates

**Proposition C.1.** *In the limit where $\lambda \to 0$ ( in equation 45), $\hat{h}_n = h_n^{(b)} + W_n^+ \Delta h_{n+1}^{(b)}$, which is a Gauss-Newton update on initial activities $h_n^{(b)}$ with residual $\Delta h_{n+1}^{(b)}$ and Jacobian $J = W_n$.*

*Proof.*

$$
\begin{aligned}
\lim_{\lambda\to 0}\hat{h}_n^{(b)} &= (W_n^T W_n)^{-1} W_n^T \hat{h}_{n+1}^{(b)} = W_n^+ \hat{h}_{n+1}^{(b)} = h_n^{(b)} + W_n^+ \hat{h}_{n+1}^{(b)} - W_n^+ h_{n+1}^{(b)} \\
&= h_n^{(b)} + W_n^+(\hat{h}_{n+1}^{(b)} - h_{n+1}^{(b)}) = h_n^{(b)} + W_n^+ \Delta h_{n+1}^{(b)},
\end{aligned}
\tag{46}
$$

$\square$

More generally, in the case where the influence of the top down error term $\|\hat{h}_n - p_n^{(b)}\|^2$ is negligible, G-IL targets converge to Gauss-Newton targets.

## C.4   Gauss-Newton G-IL as a Closed Form Approximation of G-IL

In practice, one approximates $\operatorname{argmin}_{\hat{h}_n} F$ by initializing $h_n = \hat{h}_n = p_n$ then minimizes $F$ using some optimization process. At the beginning of this optimization process the top down error $\|\hat{h}_n - p_n^{(b)}\|^2$ is small since it is initialized to zero and grows slowly afterward. Additionally, if optimization is stopped early (which is typically done in practice) this error's effect on activities will remain negligible compared to the larger bottom-up error term. This fact along with proposition C.1 suggest Gauss-Newton updated activities are a good approximation of G-IL local targets. Let's define a network that computes $\hat{h}_n$ using Gauss-Newton updates as follows

**Definition C.1.** *Gauss-Newton G-IL (IL-GN). IL-GN is a closed-form approximation of G-IL for training MLPs which uses the same weight update as G-IL and computes local targets using a Gauss-Newton update. In linear networks the update is $\hat{h}_n = h_n^{(b)} + \gamma W_n^+ \Delta h_{n+1}^{(b)}$, where $0 < \gamma \le 1$.*

IL-GN approximates G-IL when $\gamma = 1$ as shown by proposition 46. A $\gamma < 1$, however, better captures the fact that in the true closed form solution (see equation 45), $\hat{h}_n$ is a value in between a regularized GN-target and top-down prediction. Simulations below provide further justification that IL-GN with $0 < \gamma < 1$ is a good approximation of G-IL (see figure 7.

## C.5   Some Properties of IL-GN

There are several lemmas concerning IL-GN that will either be used in subsequent proofs or are relevant for understanding IL generally.

The first lemma says that predictions $p_n$ in IL-GN networks are a weighted average of $h_n$ and $\hat{h}_n$.

**Lemma C.1.** *After all $\hat{h}_n$ are computed, local predictions will lie between local sub-targets and FF activity: $p_n = (1 - \gamma)h_n + \gamma\hat{h}_n$.*

*Proof.* First, prediction $p_n$ can be described as follows: $p_n = W_{n-1}(\hat{h}_{n-1} + \Delta\hat{h}_{n-1}) = W_{n-1}(h_{n-1} + \gamma W_{n-1}^+\Delta h_n)$, which implies

$$
\begin{aligned}
p_n &= W_{n-1}(h_{n-1} + \gamma W_{n-1}^+\Delta h_n) \\
&= h_n + \gamma(\hat{h}_n - h_n) \\
&= (1 - \gamma)h_n + \gamma\hat{h}_n.
\end{aligned}
\tag{47}
$$

$\square$

It then follows that the error after the targets and predictions are updated, $e_n$, is smaller but proportional to the initial error:

**Lemma C.2.** *After all $\hat{h}_n$ are computed, local errors $e_n = (1 - \gamma)\Delta h_n$.*

*Proof.*

$$
\begin{aligned}
e_n &= \hat{h}_n - p_n \\
&= \hat{h}_n - ((1 - \gamma)h_n + \gamma\hat{h}_n) \\
&= (1 - \gamma)(\hat{h}_n - h_n) \\
&= (1 - \gamma)\Delta h_n,
\end{aligned}
\tag{48}
$$

where the second line is computed using 47. $\square$

**Lemma C.3.** *After all $\hat{h}_n$ are computed, local errors $e_n = (1 - \gamma)\gamma W_n^+\Delta h_{n+1}$.*

*Proof.*

$$
\begin{aligned}
e_n &= (1 - \gamma)\Delta h_n \\
&= \gamma(1 - \gamma)W^+\Delta h_{n+1} \\
&= \gamma W^+ e_{n+1},
\end{aligned}
\tag{49}
$$

where the first and third lines are computed using lemma C.2 and the second line using definition C.1. $\square$

These lemmas can be used to show local errors are smaller than but proportional to the global loss GN update w.r.t. hidden layer activities. For example, $e_n$ can be expressed as follows:

**Proposition C.2.** *Consider a linear MLP trained with IL-GN. Assume $(W_n^+...W_{N-1}^+) = (W_{N-1}...W_n)^+$ for all $n$. In this case, $e_n = \gamma_n'(W_{N-1}...W_n)^+\frac{-\partial L}{\partial h_N}$, where $\gamma_n' = \gamma^{(N-n)}(1 - \gamma)$.*

*Proof.* According to definition C.1, $\hat{h}_n = h_n + \gamma W_n^+\Delta h_{n+1}$, which implies $\Delta h_n = \gamma W_n^+\Delta h_{n+1}$. Applying this same operation recursively from the output layer backward we get $\Delta h_n = \gamma^{N-n}(W_n^+...W_{N-1}^+)\Delta h_N$.

Now we substitute $\gamma^{N-n}(W_n^+...W_{N-1}^+)\Delta h_N$ for $\hat{h}_n$ in lemma C.2, which results in $e_n = \gamma^{N-n}(1 - \gamma)(W_{N-1}...W_n)^+\Delta h_N$. Finally, we note that $\Delta h_N = \frac{-\partial L}{\partial h_N}$, and thus $e_n = \gamma^{(N-n)}(1 - \gamma)(W_{N-1}...W_n)^+\frac{-\partial L}{\partial h_N}$. $\square$

Thus, local errors $e_n$ can be seen as scaled down version of the global GN update, which is $\Delta h_n(W_{N-1}...W_n)^+\frac{-\partial L}{\partial h_N} = J_{n,N}^+\frac{-\partial L}{\partial h_N}$. The scaling is such that the nearer $e_n$ is to the input layer the more scaled down the error is.

# D   Stability of IL versus BP

In this section, we analyze certain stability properties of IL and BP algorithms. We begin by briefly discussing the unconditional stability of IL-prox, and equivalent G-IL algorithms. These algorithms use the NLMS update rule, so we next analyze the case where the LMS rule is used in an IL algorithm to update weights, as this case is more easily compared to G-BP algorithms which perform and LMS update over weights. In particular, we analyze how the FF output layer values $h_N$ change after weight updates are applied to a linear network using IL-GN with the LMS rule and using an analogous BP algorithm, which we call BP-GN. We show IL-GN, trained with the LMS rule, pushes output layer activities $\hat{h}_N$ toward the target $y$ (down it loss gradient) *for any positive learning rate*, while BP-GN only does this for a small finite range of learning rates. We focus on linear networks because constraining neuron activities with non-linearites may improve the stability of an algorithm by limiting the possible values a neuron may take or, e.g., by decreasing the magnitude of weight updates. We want to separate the effects of learning rules/algorithms on stability from those imposed by architectural constraints, so we focus on linear networks.

These theorems show that the way IL-GN computes and uses targets in its weight updates is a key mechanism that aids stability. In particular, IL-GN computes weight gradients for $W_n$ using pre-synaptic target $\hat{h}_n$, while BP-GN uses pre-synaptic FF activities $h_n$. This is the main difference between the two algorithms. The LMS update in the IL algorithm in linear networks is $e_{n+1}\hat{h}_n^T = (\hat{h}_{n+1} - W_n\hat{h}_n^T)\hat{h}_n^T$, while the BP update is $e_{n+1}h_n^T = (\hat{h}_{n+1} - W_nh_n)h_n^T$. IL updates thus 'chain together' the local learning problems such that the local prediction target at one layer, is the input to the next local prediction problem. G-BP updates do not have this property. The input to one local prediction problem is the FF activity, which is computed independently from local prediction targets. These results show that the way IL chains together the local prediction problems plays an important role in its stability. In particular, it shows it plays an important role in ensuring that output layer FF activities $h_N$ change in the desired *direction* across a large range of learning rates. If we update weights like BP and do not chain the local learning problems together in this way, we lose this stability guarantee.

Ensuring $h_N$ changes in the desired direction for any learning does not guarantee the loss will be minimized, since, e.g., $h_N$ may overshoot the global target significantly. As we explain next, however, one can ensure minimization across any positive learning rate using the strategy of IL-prox, which uses the NLMS rule and uses the learning rate to control the output layer target rather than to scale the weight updates.

## D.1   The Unconditional Stability of IL-prox

Implicit SGD is *unconditionally stable* in the sense that the proximal algorithm (equation 12), which is equivalent to performing an implicit SGD update, monotonically decreases the loss function $L$ for $0 < \alpha$. This can be seen from the proximal update, equation 12, from which it trivially follows $L(\theta^{(b+1)}) + \frac{1}{\alpha 2}\|\theta^{(b+1)} - \theta^{(b)}\|^2 \leq L(\theta^{(b)})$. Theorem B.2 shows that IL-prox and G-IL, under the $\gamma$ settings specified in theorem 4.1, are equivalent to an implicit SGD update, and thus are unconditionally stable. Our simulations find that our implementation of IL-prox indeed displays this property of unconditional stability (tables 5 and 4). It is worth, however, briefly discussing the mechanics of how this unconditional stability comes about in the IL-prox algorithm (and equivalent G-IL algorithms). First, theorem 4.1, tells us that the learning rate $\alpha$ of IL-prox (and G-IL under certain $\gamma$ settings), is only used to determine how much $\hat{h}_N$ is pushed toward global target $y$ during the inference phase. More specifically the gradient of the proximal update w.r.t. $\hat{h}_N$ is

$$\frac{\partial Prox}{\partial \hat{h}_N} = \frac{\partial L}{\partial \hat{h}_N} + \frac{1}{\|\hat{h}_{N-1}\|^2\alpha}e_N, \tag{50}$$

where $e_N = \hat{h}_N - p_N$ and $p_N = W_{N-1}f(\hat{h}_{N-1})$. Let's assume a linear network and $L = \frac{1}{2}\|y - \hat{h}_N\|^2$. One way to compute the update of $\hat{h}_N$ would be to take the negative of the gradient,

set equal to zero and solve for $\hat{h}_N$:

$$0 = -(y - \hat{h}_N) - \frac{1}{\|\hat{h}_{N-1}\|^2 \alpha} e_N,$$

$$\hat{h}_N = \frac{\|\hat{h}_{N-1}\|^2 \alpha}{1 + \|\hat{h}_{N-1}\|^2 \alpha} y + \frac{1}{1 + \|\hat{h}_{N-1}\|^2 \alpha} p_N. \tag{51}$$

Thus, each iteration of the inference phase we update $\hat{h}_N$ to a weighted average between $p_N$ and $y$. The learning rate is only used to determine the weighting between these two terms. As $\alpha \to \infty$, $\hat{h}_N \to y$ and as $\alpha \to 0$, $\hat{h}_N \to p_N$. According to lemma B.1, $\hat{h}_N$ becomes the FF output layer value after the weights are updated (since the NLMS rule is used by IL-prox). Thus, when $\alpha \to \infty$, IL-prox finds a set of weights that best minimize the loss and as $\alpha \to 0$, weight are unchanged, which is the exact same property the proximal algorithm has (equation 12). Intuitively, we see that in any case the loss is either minimized or remains unchanged (when $\alpha = 0$). IL-prox, and the proximal algorithm, thus gain their stability from the fact they do not use the learning rate to scale weight updates (as explicit SGD does) but rather uses it to determine the relative weighting of the loss term and regularization term in the proximal operator.

## D.2 Gauss-Newton IL Updates are Minimum-Norm and Stable

Here we show that the IL-GN weight updates collectively minimizes global loss along its minimum norm path for any positive learning rate in linear networks. This theorem is proven true under the conditions where $\hat{h}_n^T p_n > 0$, $h_n^T h_n > 0$, and $\hat{h}_n^T \hat{h}_n > \hat{h}_n^T p_n$ for all $n$. This means local predictions, $p_n$, and initial activities, $h_n$ must be within 90 degrees of sub-targets. This is an easily satisfied condition as long as targets are not moved far from initial activities. Also, the inequality $\hat{h}_n^T \hat{h}_n > \hat{h}_n^T p_n$ is generally true since the magnitudes of $\hat{h}_n$ and $p_n$ will be similar and $\hat{h}_n$ is always more similar to itself than $p_n$. Thus, these conditions generally hold in practice.

**Theorem D.1.** *Assume $\hat{h}_n^T p_n > 0$, $\hat{h}_n^T h_n > 0$, and $\hat{h}_n^T \hat{h}_n > \hat{h}_n^T p_n$ for all $n$. For a mini-batch of size 1, the IL-GN weight updates applied to a linear MLP at iteration $b$ collectively push $h_N$ down its global loss gradient toward the target for any positive learning rate $\alpha$: $h_N^{(b+1)} = h_N^{(b)} - j^{(b)} \frac{\partial L^{(b)}}{\partial h_N^{(b)}}$ where $j^{(b)}$ is a positive scalar.*

*Proof.* We define the linear network after weight updates are applied using a recursive formulation. Let $\hat{W}_n^* = \hat{W}_{n+1}^*(W_n - \Delta W_n)$ for all $n < N - 1$, and let $\hat{W}_{N-1}^* = W_{N-1} - \Delta W_{N-1}$. We can now express the output of this updated network given input $h_0$ as $\hat{W}_0^* h_0$. We assume a linear gradient update $\Delta W_n = -\alpha e_{n+1} \hat{h}_n^T$, where $\alpha$ is the learning rate and $e_{n+1} = \hat{h}_{n+1} - p_{n+1}$.

The feedforward pass of the updated MLP at iteration $b + 1$ is described as follows:

$$h_N^{(b+1)} = \hat{W}_0^* h_0^{(b)} = \hat{W}_1^*(W_0 - \Delta W_0)h_0^{(b)}$$
$$= \hat{W}_1^* h_1^{(b)} + \alpha \hat{W}_1^* e_1^{(b)} \hat{h}_0^{T(b)} h_0^{(b)} \tag{52}$$

Note that $\hat{h}_0 = h_0$ and is the same across all training iterations, but $\hat{h}_n \neq h_n$ for $n > 0$.

Let $c_n = \alpha \hat{h}_{n-1}^T h_{n-1}$ such that $\alpha \hat{W}_1^* e_1 \hat{h}_0^T h_0 = c_1 \hat{W}_1^* e_1$. Because $\alpha > 0$ and because $\hat{h}_n^T h_n > 0$ for all $n$, $c_n > 0$.

Notice that the first term $\hat{W}_1^* h_1$ can be expanded recursively using the same expansion of $\hat{W}_0^* h_0$ in equation 52:

$$h_N^{(b+1)} = \hat{W}_1^* h_1^{(b)} + c_1 \hat{W}_1^* e_1^{(b)}$$
$$= \hat{W}_2^* h_2^{(b)} + c_1 \hat{W}_2^* e_2^{(b)} + c_1 \hat{W}_1^* e_1^{(b)}$$
$$...$$
$$= h_N + c_N e_N^{(b)} + c_{N-1} \hat{W}_{N-1}^* e_{N-1}^{(b)} + ... + c_2 \hat{W}_2^* e_2^{(b)} + c_1 \hat{W}_1^* e_1^{(b)} \tag{53}$$

The leftmost terms $h_N + c_N e_N$ was computed as follows: $(W_{N-1} - \Delta W_{N-1})h_{N-1} = h_N + \alpha e_N \hat{h}_{N-1}^T h_{N-1} = h_N + c_N e_N$.

Now we need to expand $\hat{W}_n^*$ for all $n$ in equation 53.

$$
\begin{aligned}
\hat{W}_n^* e_n^{(b)} &= \hat{W}_{n+1}^*(W_n - \Delta W_n)e_n \\
&= \hat{W}_{n+1}^*(W_n e_n - (\Delta W_n \hat{h}_n - \Delta W_n p_n)) \\
&= \hat{W}_{n+1}^*(W_n e_n + \alpha(e_{n+1}\hat{h}_n^T \hat{h}_n - e_{n+1}\hat{h}_n^T p_n)) \\
&= \hat{W}_{n+1}^*(W_n e_n + e_{n+1}\alpha(\hat{h}_n^T \hat{h}_n - \hat{h}_n^T p_n)) \\
&= \hat{W}_{n+1}^*(W_n e_n + k_n e_{n+1}),
\end{aligned}
\tag{54}
$$

Where $k_n = \alpha(\hat{h}_n^T \hat{h}_n - \hat{h}_n^T p_n)$ and is a positive scalar, since we assume $\hat{h}_n^T \hat{h}_n > \hat{h}_n^T p_n$.

We now simplify using lemma C.3, which states $e_n = \gamma W^+ e_{n+1}$:

$$
\begin{aligned}
\hat{W}_n^* e_n^{(b)} &= \hat{W}_{n+1}^*(\gamma W_n W_n^+ e_{n+1}^{(b)} + k_n e_{n+1}^{(b)}) \\
&= d_n \hat{W}_{n+1}^* e_{n+1}^{(b)},
\end{aligned}
\tag{55}
$$

where $d_n = k_n + \gamma$ and clearly $d_n > 0$.

We now apply the same derivation of $\hat{W}_n^* e_n^{(b)}$ to all $\hat{W}$ in equation 53 to yield the following:

$$
\begin{aligned}
\hat{W}_n^* e_n^{(b)} &= (d_n d_{n+1}...d_{N-2})\hat{W}_{N-1}^* e_{N-1}^{(b)} \\
&= (\prod_{i=n}^{N-2} d_i)(W_{N-1} - \Delta W_{N-1})e_{N-1}^{(b)} \\
&= (\prod_{i=n}^{N-2} d_i)(W_{N-1}e_{N-1}^{(b)} + \alpha e_N \hat{h}_{N-1}^{T(b)} e_{N-1}^{(b)}) \\
&= (\prod_{i=n}^{N-2} d_i)(W_{N-1}e_{N-1}^{(b)} + \alpha e_N (\hat{h}_{N-1}^{T(b)}\hat{h}_{N-1}^{(b)} - \hat{h}_{N-1}^{T(b)}p_{N-1}^{(b)}) \\
&= (\prod_{i=n}^{N-2} d_i)((1 - \gamma)W_{N-1}W_{N-1}^+ e_N^{(b)} + k_{N-1}e_N^{(b)}) \\
&= (\prod_{i=n}^{N-1} d_i)e_N^{(b)}
\end{aligned}
\tag{56}
$$

Note that $(\prod_{i=n}^{N-1} d_i)$ is a positive scalar since it is the product of positive scalars.

According to lemma C.2 $e_N = (1 - \gamma)\Delta h_N$ where we assume $\Delta h_N = -\frac{\partial L}{\partial h_N}$. Let $g_n = (1 - \gamma)(\prod_{i=n}^{N-1} d_i)$, and notice $g_n > 0$. We can now rewrite equation 53 in terms of $\frac{L}{h_N}$:

$$
\begin{aligned}
h_N^{(b+1)} &= h_N + c_N e_N^{(b)} + c_{N-1}g_{N-1}e_N^{(b)} + ... + c_2 g_2 e_N^{(b)} + c_1 g_1 e_N^{(b)} \\
&= h_N - j\frac{\partial L}{\partial h_N}^{(b)},
\end{aligned}
\tag{57}
$$

where $j = c_N + \sum_{i=n}^{N-1} c_i g_i$ and $j > 0$ is both $c > 0$ and $g > 0$. Hence, the weight updates applied at iteration $b$ will collectively move the FF output layer values $h_N$ down its loss gradient $h_N - j e_N^{(b)}$ for any positive value of $\alpha$. $\square$

It should be noted that moving $h_N$ down its loss gradient will not necessarily minimize the loss $e_N$, given that $j$ may be large and could cause $h_N$ to significantly overshoot $\hat{h}_N$ preventing convergence.

## D.3 Gauss-Newton Back Propagation is Minimum-Norm but Only Conditionally Stable

In this section, we show that, unlike IL-GN, weight updates performed by an analogous BP based network, we call BP-GN, are not guaranteed to collectively move the global prediction down its loss gradient each training iteration. Meulemans et al. [24] showed that each individual weight update in a BP-GN network will push the global prediction $h_N$ down its loss gradient when its effects on $h_N$ are considered independently of other weight updates in the network. Here we show this property does not hold for any learning rate when the weight updates are considered collectively.

**Definition D.1.** *Gauss-Newton backpropagation (BP-GN). Assume $e_n = \hat{h}_n - h_n$. The weight update of BP-GN is the general BP weight update: $\Delta W_n = -e_{n+1} h_n^T$. Local targets are computed at each hidden layer are computed using the following equation: $\hat{h}_n = h_n + W_n^+ e_{n+1} = h_n + W_n^+ \hat{h}_{n+1} - W_n^+ h_{n+1}$.*

We can now apply the same analysis we did for IL-GN to BP-GN, which proves the following.

**Theorem D.2.** *Consider a linear MLP trained with BP-GN. For a mini-batch of size 1, the BP-GN weight updates applied at iteration $b$ only push the global prediction down its global loss gradient $h_N^{(b+1)} = h_N^{(b)} - j^{(b)} \frac{\partial L^{(b)}}{\partial h_N^{*(b)}}$ for a finite range of $\alpha$.*

*Proof.* Here we follow the same notation and analysis as the proof for theorem D.1. Let $\hat{W}_n^* = \hat{W}_{n+1}^*(W_n - \Delta W_n)$ for all $n < N - 1$, and let $\hat{W}_{N-1}^* = W_{N-1} - \Delta W_{N-1}$. We assume a linear gradient update $\Delta W_n = -\alpha e_{n+1} h^T$ (as in definition **??**), where $\alpha$ is the learning rate.

First, we can describe the FF pass of the updated MLP trained with GN-TP at iteration $b + 1$ as we did above for IL-GN (see equation 52):

$$h_N^{(b+1)} = \hat{W}_1^* h_1^{(b)} + \alpha \hat{W}_1^* e_1^{(b)} h_0^{T(b)} h_0^{(b)} \tag{58}$$

Let $c_n = \alpha h_{n-1}^T h_{n-1}$ such that $\alpha \hat{W}_1^* e_1 h_0^T h_0 = c_1 \hat{W}_1^* e_1$. Because $1 > \alpha > 0$ and because $h_n^T h_n > 0$ for all $n$, $c_n > 0$. We can see this equation is the same as equation 52, except $e_1$ and $\Delta W_0$ were defined according to GN-TP rather than IL-GN.

As above, the first term $\hat{W}_1^* h_1$ can be expanded recursively:

$$\begin{aligned}
h_N^{(b+1)} &= \hat{W}_1^* h_1^{(b)} + c_1 \hat{W}_1^* e_1^{(b)} \\
&= \hat{W}_2^* h_2^{(b)} + c_1 \hat{W}_2^* e_2^{(b)} + c_1 \hat{W}_1^* e_1^{(b)} \\
&\cdots \\
&= h_N - c_N e_N^{(b)} + c_{N-1} \hat{W}_{N-1}^* e_{N-1}^{(b)} + \dots + c_2 \hat{W}_2^* e_2^{(b)} + c_1 \hat{W}_1^* e_1^{(b)}
\end{aligned} \tag{59}$$

Equation 53 shows that for IL-GN, each of these recursively computed terms push the initial prediction down the global loss gradient. However, the same is not true here of GN-TP:

$$\begin{aligned}
\hat{W}_n^* e_n^{(b)} &= \hat{W}_{n+1}^* (W_n - \Delta W_n) e_n \\
&= \hat{W}_{n+1}^* (W_n e_n - (\Delta W_n \hat{h}_n - \Delta W_n h_n)) \\
&= \hat{W}_{n+1}^* (W_n e_n + \alpha(e_{n+1} h_n^T \hat{h}_n - e_{n+1} h_n^T h_n)) \\
&= \hat{W}_{n+1}^* (W_n e_n + \alpha(h_n^T \hat{h}_n - h_n^T h_n) e_{n+1}).
\end{aligned} \tag{60}$$

$W_n e_n$ can be rewritten in terms of $e_{n+1}$ as follows: $W_n e_n = W_n W_n^+ e_{n+1}$. Notice that it will often be the case that $h_n^T \hat{h}_n < h_n^T h_n$, since generally $h_n \neq h_n$. Thus, $\alpha(h_n^T \hat{h}_n - h_n^T h_n)$ will often be a *negative* scalar. This is distinct from the corresponding term in equation for IL-GN $\alpha(h_n^T h_n - \hat{h}_n^T h_n)$, which will generally be a positive scalar. The difference is due to the IL-GN update being conditioned

on the presynaptic sub-target $\hat{h}_n$ rather than presynaptic FF activity $h_n$. Also notice there is no further dampening term $\gamma(1 - \gamma)$ as there is with IL-GN.

Let $k_n = \alpha(h_n^T \hat{h}_n - h_n^T h_n)$ and $d_n = (1 + k_n)$. For reasons just noted, $d_n$ is not guaranteed to be positive for all $1 > \alpha > 0$. Continuing the derivation above:

$$
\begin{aligned}
\hat{W}_n^* e_n^{(b)} &= \hat{W}_{n+1}^*(W_n W_n^+ e_{n+1} + k_n e_{n+1}) \\
&= \hat{W}_{n+1}^*(1 + k_n)e_{n+1} \\
&= d_n \hat{W}_{n+1}^* e_{n+1}^{(b)} \\
&= (d_n d_{n+1}...d_{N-2})\hat{W}_{N-1}^* e_{N-1}^{(b)} \\
&= (\prod_{i=n}^{N-1} d_i)e_N^{(b)}.
\end{aligned}
\tag{61}
$$

Because $d_n$ is not guaranteed to be positive, then $(\prod_{i=n}^{N-1} d_i)$ is not guaranteed to be a positive scalar. Thus, the product of all $d_n$ may or may not be positive for a given $1 > \alpha > 0$. Let $g_n = (\prod_{i=n}^{N-1} d_i)$. Also, note that $e_N = -\frac{\partial L}{\partial h_N}$. We can now rewrite equation 58 in terms of $e_N$:

$$
\begin{aligned}
h_N^{(b+1)} &= h_N + c_N e_N^{(b)} + c_{N-1}g_{N-1}e_N^{(b)} + ... + c_2 g_2 e_N^{(b)} + c_1 g_1 e_N^{(b)} \\
&= h_N - j\frac{\partial L}{\partial h_N}^{(b)},
\end{aligned}
\tag{62}
$$

We can see that if enough $g_n$ are negative it is possible that $j$ is also negative, especially given that $g_n$ are scaled by $c_n$, which will always be positive and may be larger than 1. Thus, GN-TP, unlike IL-GN, does not guarantee that for any positive value of $\alpha$, the weight updates at iteration $b$ will collectively move $h_N$ down it loss gradient at $b + 1$: GN-TP does not guarantee $j > 0$ for any $\alpha > 0$ or even any $1 > \alpha > 0$. $\qquad \square$

In order for $j > 0$, a large number of $g_n$ must be positive to offset any negative $g_n$. In order for $g_n > 0$, an even number of $d_n$ in $g_n = (\prod_{i=n}^{N-1} d_i)$ must be negative. The simplest way to guarantee this is the case is to ensure $d_n > 0$ for all $n$. In order for all $d_n$ to be positive in each $g_n$ we can see that alpha must be small since $d_n = 1 + k_n = 1 + \alpha(h_n^T \hat{h}_n - h_n^T h_n)$, and $\alpha$ must be small enough such that $k_n > -1$ for all $n$. This implies learning rate must be small: $\alpha < \frac{-1}{(h_n^T \hat{h}_n - h_n^T h_n)}$.

## E  G-IL Avoids Interference Effects

The last section showed the IL-GN will push output layer values toward the global target for any positive learning rate, while BP-GN will only do so for a finite range of learning rates. Here we provide mathematical intuition for why this occurs. Specifically, we describe how BP weight updates necessarily interfere with each other's ability to minimize the global loss and this interference generally grows with learning rate, whereas IL weight updates do not necessarily interference and may actually improve each other's ability to minimize loss. In this sense, IL updates are more compatible with each other than BP updates. In line with this intuitive description, we show that under certain assumptions IL-GN updates (at hidden layers) are often *most effective* at pushing the output layer values toward the target when applied in conjunction with other updates than when applied alone, while the opposite is true of BP-GN. Figure 8 provides evidence these theoretical results hold true in practice for IL-SGD and BP-SGD networks when small learning rates are used.

### E.1  A Mathematical Intuition for Interference Effects

Consider the BP weight update: $\Delta W_n = \alpha_n e_{n+1}^{(b)} f(h_n)^{T(b)}$, where $h_n = W_{n-1}f(h_{n-1})$ and $n > 0$. Now, consider the effect of this weight update on the FF values at hidden layer $n + 1$ given the same

input $x^{(b+1)} = x^{(b)}$

$$
\begin{aligned}
h_{n+1}^{(b+1)} = W_n^{(b+1)} f(h_n^{(b+1)}) &= (\Delta W_n^{(b)} + W_n^{(b)}) f(h_n^{(b+1)}) \\
&= \Delta W_n^{(b)} f(h_n^{(b+1)}) + W_n^{(b)} f(h_n^{(b+1)}) \\
&= \alpha_n (f(h_n^{(b)})^T f(h^{(b+1)})_n)) e_{n+1}^{(b)} + h_{n+1}^{(b)}.
\end{aligned}
\tag{63}
$$

We can see that the ability of the weight update to drive its output down the error gradient (where $e_{n+1}^{(b)} = \frac{-\partial L}{\partial h_{n+1}^{(b)}}$, see section 3.2) depends directly on the similarity between pre-synaptic activities before and after the weight update. In particular, if the two are similar, i.e. $f(h_n^{(b)})^T f(h_n^{(b+1)}) > 0$, then the weight update will affect FF values in the desired direction. However, if the two are orthogonal, i.e. $f(h_n^{(b)})^T f(h_n^{(b+1)}) = 0$, there will be no effect, and if dissimilar, i.e. $f(h_n^{(b)})^T f(h_n^{(b+1)}) < 0$, they will affect FF values in an undesired direction (up rather down the loss gradient).

The challenge for the BP update is that if all weights are updated at iteration $b$ then FF activities will change, which will generally make $f(h_n^{(b)})$ and $f(h_n^{(b+1)})$ less similar. Additionally, presynaptic activities will generally become less similar as $\alpha_n$ is increased because changes to weights, and thus changes to FF activities, at previous layers will increase. Only when learning rates at previous layers $\to 0$ does $f(h_n)^{(b)} = f(h_n)^{(b+1)}$ because an $\alpha$ of zero means no change in weights or FF activities. We can describe this as a sort of interference effect: non-zero weight updates elsewhere in the network reduce the ability of $\Delta W_n$ to drive FF activities at layer $n + 1$ in the desired direction.[2]

Now, consider the G-IL weight update: $\Delta W_n = \alpha_n e_{n+1}^{(b)} f(\hat{h}_n^{T(b)})$, where $n > 0$. The effect of this weight update on the FF values at hidden layer $n + 1$ given the same input, $x^{(b+1)} = x^{(b)}$, can be described as follows:

$$
\begin{aligned}
h_{n+1}^{(b+1)} = W_n^{(b+1)} f(h_n^{(b+1)}) &= (\Delta W_n^{(b)} + W_n^{(b)}) f(h_n^{(b+1)}) \\
&= \Delta W_n^{(b)} f(h_n^{(b+1)}) + W_n^{(b)} f(h_n^{(b+1)}) \\
&= \alpha_n (f(\hat{h}_n)^{T(b)} f(h_n^{(b+1)})) e_{n+1}^{(b)} + h_{n+1}^{(b)}.
\end{aligned}
\tag{64}
$$

Unlike with G-BP we can see that the ability of the weight update to drive its output down the error gradient $e_{n+1}^{(b)}$ depends directly on the similarity between pre-synaptic *optimized* activities and the FF activities after the weight update. If the two are similar, i.e. $f(\hat{h}_n^{T(b)}) f(h_n^{(b+1)}) > 0$, then the weight update will affect FF values in the desired direction. If the two are orthogonal there will be no effect, and if they are dissimilar they will affect FF values in an undesired direction.

What's interesting about the G-IL update is that with a properly tuned $\alpha_n$, $f(\hat{h}_n)^{(b)}$ and $f(h_n)^{(b+1)}$ will get *more* similar when weights prior to layer $n$ are updated. As lemma B.1 shows, under a specific (typically) non-zero value of $\alpha_n$ it is actually the case that $f(\hat{h}_n^{(b)}) = f(h_n^{(b+1)})$. This value of $\alpha_n$ is the true solution to G-IL update $\operatorname{argmin}_W F$ that the LMS learning rule above attempts to approximate. Intuitively, G-IL largely avoids interference because $f(\hat{h}_n^{(b)})$ accurately *anticipates* the value of $f(h_n^{(b+1)})$ and by doing so helps to ensure $f(\hat{h}_n)^{T(b)} f(h_n)^{(b+1)} > 0$.

### E.2 G-IL Weight Updates are Most Effective when Applied Together

In previous sections we compared the direction of the effects of weight updates on a linear MLP output layer in an IL and BP network. We now assess the magnitude of the affects of IL and BP weight updates on the output layer of a linear MLP with a single hidden layer. In sum, we show that under certain assumptions, updates in IL-GN should have greater effects on the output layer when applied together than when applied alone, while the opposite is true for BP-GN. This shows that interference effects not only can alter the direction the output layer changes with weight updates but also the magnitude of the change in output layer activities.

---

[2]Interference might be reduced by using non-linearities that keep activities positive, e.g., ReLU, which guarantees $f(h_n^{T(b)}) f(h_n^{(b+1)}) \geq 0$. However, for non-zero learning rates it would still be the case that updates to weights at layers before $n$ would reduce similarity, i.e., generally $f(h_n^{T(b)}) f(h_n^{(b)}) > f(h_n^{T(b)}) f(h_n^{(b+1)})$, and thus reduce the magnitude change to post-synaptic FF values.

We analyze a linear MLP with a single hidden layer. Let $h_2^{(b)}$ be the output layer values at iteration $b$, $h_2^{(b+1/2),0}$ be the output layer values after $W_0$ is updated alone, $h_2^{(b+1/2),1}$ be the output layer values after $W_1$ is updated alone, and $h_2^{(b+1)}$ be the output layer values after both weight matrices are updated. Let $c_n$ and $g_n$ be defined as above in the proof of theorem D.1. We assume that $g_n > 1$ which will often (though not always) be true in practice for reasons noted above (see theorem D.1). We also assume $c_1 = c_2$ as it significantly simplifies the calculations. We test how well these theoretical results hold in practice and find they hold approximately for small learning rates (see figures 8 for details).

**Proposition E.1.** *Consider a linear MLP with a single hidden layer trained with IL-GN, mini-batch size 1, at iteration $b$. Assume $g_1 > 1$, $c_1 = c_2$. Under these assumptions, and those in theorem D.1, each weight update, $\Delta W_0$ and $\Delta W_1$ has a larger effect on the output layer when applied in conjunction with each other than when applied alone: $\|h_2^{(b+1)} - h_2^{(b+1/2),0}\| > \|h_2^{(b+1/2),1} - h_2^{(b)}\|$ and $\|h_2^{(b+1)} - h_2^{(b+1/2),1}\| > \|h_2^{(b+1/2),0} - h_2^{(b)}\|$*

*Proof.* The effect of $\Delta W_0$ on the output layer $h_N = h_2$ when applied alone is

$$
\begin{aligned}
h_2^{(b+1/2),0} = W_1^{(b)}(W_0^{(b)} + \Delta W_0^{(b)})h_0^{(b)} &= h_2^{(b)} + W_1^{(b)}\alpha e_1^{(b)} h_0^{(b)T} h_0^{(b)} \\
&= h_2^{(b)} + \alpha\gamma W_1^{(b)} W_1^{(b)+} e_2^{(b)} h_0^{(b)T} h_0^{(b)} \\
&= h_2^{(b)} + \alpha\gamma h_0^{(b)T} h_0^{(b)} e_2^{(b)} \\
&= h_2^{(b)} + \gamma c_1 e_2^{(b)},
\end{aligned}
\tag{65}
$$

where the second line is computed using C.3, which says $e_n = \gamma W_n^+ e_{n+1}$. Here $c_1 = \alpha h_0^{(b)T} h_0^{(b)}$ and is clearly a positive scalar.

The effect of $\Delta W_1$ on the output layer, $h_2$, when applied alone is

$$
\begin{aligned}
h_2^{(b+1/2),1} = (W_1^{(b)} + \Delta W_1^{(b)})W_0^{(b)} h_0^{(b)} &= h_2^{(b)} + \Delta W_1^{(b)} h_1^{(b)} \\
&= h_2^{(b)} + \alpha \hat{h}_1^{(b)T} h_1^{(b)} e_2^{(b)} = h_2^{(b)} + c_2 e_2^{(b)},
\end{aligned}
\tag{66}
$$

where $c_2 = \alpha \hat{h}_1^{(b)T} h_1^{(b)}$ and is a scalar.

Finally, when both updates are applied we get the same expression as that in equation 57

$$
h_2^{(b+1)} = h_2^{(b)} + c_2 e_2^{(b)} + g_1 c_1 e_2^{(b)}.
\tag{67}
$$

See theorem D.1 for details. Here $c_1$ and $c_2$ are defined as above. Since we make the same assumptions as theorem D.1, $g_1$ is a positive scalar: $g_1 = \gamma + \alpha(\hat{h}_1^{(b)T} \hat{h}_1^{(b)} - \hat{h}_1^{(b)T} p_1^{(b)})$.

Now we have

$$
\|h_2^{(b+1)} - h_2^{(b+1/2),0}\| = \|c_2 e_2^{(b)} + g_1 c_1 e_2^{(b)} - \gamma c_1 e_2^{(b)}\| = \|(1 + g_1 - \gamma)c_1 e_2^{(b)}\|,
\tag{68}
$$

which is simplifying using the assumption $c_1 = c_2$. Second, $\|h_2^{(b+1/2),1} - h_2^{(b)}\| = \|c_2 e_2^{(b)}\|$. Under our assumptions that $g_1 > 1$ and $c_1 = c_2$, it clearly follows that $\|h_2^{(b+1)} - h_2^{(b+1/2),0}\| > \|h_2^{(b+1/2),1} - h_2^{(b)}\|$.

Next, we can see that $\|h_2^{(b+1)} - h_2^{(b+1/2),1}\| = \|g_1 c_1 e_2^{(b)}\|$ and $\|h_2^{(b+1/2),0} - h_2^{(b)}\| = \|\gamma c_1 e_2^{(b)}\|$. Since we assume $g_1 > 1$ and $c_1 = c_2$ and $\gamma < 1$, it clearly follows that $\|h_2^{(b+1)} - h_2^{(b+1/2),1}\| > \|h_2^{(b+1/2),0} - h_2^{(b)}\|$. $\qquad\square$

### E.3 BP-GN Updates are Most Effective When Applied Alone

Next, we perform the same analysis on BP-GN and find the opposite is true: BP-GN updates are most effective when applied alone. We assume $g_n < 1$, which is plausible given that $g_n$ is typically a negative scalar for reasons noted in the proof for theorem D.2. We also make the same assumption as we did for IL-GN that $c_1 = c_2$.

**Proposition E.2.** *Consider a linear MLP with a single hidden layer trained with BP-GN, mini-batch size 1, at iteration $b$. Assume generally $g_n < 1$ and $c_1 = c_2$. The weight update to $\Delta W_0$ and $\Delta W_1$ either have a larger effect on the output layer when applied alone:* $\|h_2^{(b+1)} - h_2^{(b+1/2),0}\| < \|h_2^{(b+1/2),1} - h_2^{(b)}\|$ *and* $\|h_2^{(b+1)} - h_2^{(b+1/2),1}\| < \|h_2^{(b+1/2),0} - h_2^{(b)}\|$ *or they push the output layer activities up the loss gradient (i.e., away from global target).*

*Proof.* Following equation 65 above, the effect of $\Delta W_0$ on the output layer is $h_2^{(b)} + c_1 e_2^{(b)}$, except $c_1$ is now computed $c_1 = h_0^{T(b)} h_0^{(b)}$.

The effect of $\Delta W_1$ on the output layer, $h_2$, when the update is applied alone is

$$
\begin{aligned}
h_2^{(b+1/2),1} &= (W_1^{(b)} + \Delta W_1^{(b)}) W_0^{(b)} h_0^{(b)} = h_2^{(b)} + \Delta W_1^{(b)} h_1^{(b)} \\
&= h_2^{(b)} + \alpha h_1^{(b)T} h_1^{(b)} e_2^{(b)} = h_2^{(b)} + c_2 e_2^{(b)},
\end{aligned}
\tag{69}
$$

where $c_2 = \alpha h_1^{(b)T} h_1^{(b)}$ and is generally a large positive scalar.

Finally, using the derivation from theorem D.2, when both updates are applied we get

$$
h_2^{(b+1)} = h_2^{(b)} + c_2 e_2^{(b)} + c_1 g_1 e_2^{(b)}
\tag{70}
$$

where $g_1 = 1 + \alpha(h_1^{(b)T} \hat{h}_1^{(b)} - h_1^{(b)T} h_1^{(b)})$ (see theorem D.2 for details). As explained in theorem D.2, we can see that $g_1$ will often be negative since it will often be the case that $(h_1^{(b)T} \hat{h}_1^{(b)} - h_1^{(b)T} h_1^{(b)}) < 0$.

It follows from the assumption $c_1 = c_2$ that $\|h_2^{(b+1)} - h_2^{(b+1/2),0}\| = \|g_1 c_1 e_2^{(b)}\|$. Second, $\|h_2^{(b+1/2),1} - h_2^{(b)}\| = \|c_2 e_2^{(b)}\|$. Under our assumptions $g_1 < 1$ and $c_0 = c_1$, in the case where $-1 < g_1 < 1$, it follows $\|h_2^{(b+1)} - h_2^{(b+1/2),0}\| < \|h_2^{(b+1/2),1} - h_2^{(b)}\|$ and generally when $g_1 < -1$, and under assumption $c_1 = c_2$, we see from equation 70 the output layer activities away from the target (i.e. up rather than down the loss gradient).

Similarly, $\|h_2^{(b+1)} - h_2^{(b+1/2),0}\|^2 = \|g_1 c_1 e_2^{(b)}\|$ and $\|h_2^{(b+1/2),0} - h_2^{(b)}\| = \|c_1 e_2^{(b)}\|$. Under our assumptions $g_1 < 1$ and $c_1 = c_2$, in the case where $-1 < g_1 < 1$ it follows $\|h_2^{(b+1)} - h_2^{(b+1/2),0}\| < \|h_2^{(b+1/2),1} - h_2^{(b)}\|$ and again when $g_1 < -1$ it is the case the the output layer activities are pushed away from the target (i.e. up rather than down the loss gradient). $\qquad\square$

Under the above assumptions, IL-GN weight updates do not interfere with one another, and in fact are more effective when applied together than alone. BP-GN weight updates, on the other hand, tend to be most effective when applied alone. We test how well these theoretical results are approximated by IL-SGD and BP-SGD through simulations on linear networks trained on regression tasks. Indeed, IL-SGD networks tend to have larger effects on the output layer when applied together than along, more often than their BP counterparts when small learning rates are used (see figure 8).

## F   Supplementary Results and Experiments

### F.1   Cifar-10 Mini-batch Classification

In figure 5, are training runs for Cifar-10 mini-batch size 64. We see that IL prox algorithms tend to discount the loss faster than BP-SGD and are comparable to BP-Adam in the first 2000 iterations. However, BP-SGD and BP-Adam converge to better accuracies. This suggests that IL algorithms tend to converge to shallower local minima that are nearer by the initial parameters than those minima reached by BP algorithms.

### F.2   Further Stability Analysis Results and Discussion

We run the same stability analysis as in table 4 for MNIST. BP-prox is unable to train in a stable manner for learning rates in the range of 2.5-100, and there is a clear decrease in accuracy as learning

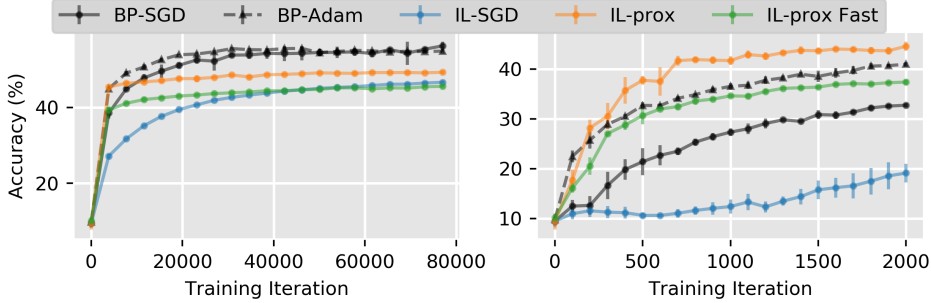

Figure 5: Cifar-10 classification test accuracies during training with mini-batch size 64. Left, full training run. Right, a zoomed in look at test accuracies from the first 2000 training iterations.

rate increases from .01 to 1. IL-prox's trains across all learning rates and even improves as learning rate increase from .01 to 1. As with the Cifar-10 results we again see IL-SGD is more stable than BP-SGD, and prox models are more stable than SGD models.

BP-prox uses the NLMS version of its update (equation 3) and uses the learning rate to adjust the output layer target. Like IL-prox, as $\alpha \to \infty$ then $\hat{h}_N \to y$ and as $\alpha \to 0$ then $\hat{h}_N \to h_N$. This use of the NLMS rule and learning rate improve stability. However, clearly these factors do not give BP-prox unconditional stability. The main difference between IL-prox and BP-prox is the way local targets are computed and used in the weight update. For example, while BP-prox uses presynaptic FF activity $h_n$ in its update of $W_n$, IL-prox uses presynaptic target value $\hat{h}_n$ in its update of $W_n$ (compare equations 75 and 77). Clearly, these differences have a significant effect on stability.

| Stability Test: MNIST Test Accuracy | | | | | | |
|---|---|---|---|---|---|---|
| Model | lr=.01 | lr=.1 | lr=1 | lr=2.5 | lr=10 | lr=100 |
| BP-SGD | 94.05($\pm$.53) | $-$ | $-$ | $-$ | $-$ | $-$ |
| IL-SGD | 90.06($\pm$.25) | 94.49($\pm$.21) | 42.42($\pm$44.67) | $-$ | $-$ | $-$ |
| BP-prox | 93.028($\pm$.77) | 90.86($\pm$1.62) | 55.87($\pm$42.076) | $-$ | $-$ | $-$ |
| IL-prox | 89.30($\pm$.32) | 93.058($\pm$.56) | 93.57($\pm$.32) | 93.49($\pm$.61) | 93.49($\pm$.94) | 93.85($\pm$.38) |

Table 5: Accuracy after 20,000 training iterations on MNIST mini-batch size 1. Fully connected networks with layer sizes 784-2x500-10. ReLU activations were used at hidden layers while softmax was used at output layer

### F.3   MNIST Training with Mini-batch Size 1

| Test Accuracy (mean$\pm$std.) w/ Best Learning Rate | | |
|---|---|---|
| Model | MNIST | Fashion-MNIST |
| BP-SGD | 97.096($\pm$.131) | 85.384($\pm$.120) |
| BP-Adam | 97.294($\pm$.116) | 85.896($\pm$.081) |
| IL-SGD | 96.938($\pm$.102) | 84.972($\pm$.211) |
| IL-prox | 96.650($\pm$.091) | 83.636($\pm$.091) |
| IL-prox Fast | 96.466($\pm$.128) | 83.402($\pm$.169) |

Table 6: Test accuracies after 1 epoch of training with mini-batch size 1 (60,000 training iterations). Fully connected networks were used with dimensions 784-2x500-10.

The training runs for are plotted in figure 6 below. Differences in performance between IL and BP algorithms are smaller with these data-sets than they are with CIFAR-10. IL-prox improved accuracy slightly faster in the first few hundred or thousand iterations on both MNIST and Fashion-MNIST, but BP algorithms soon caught up.

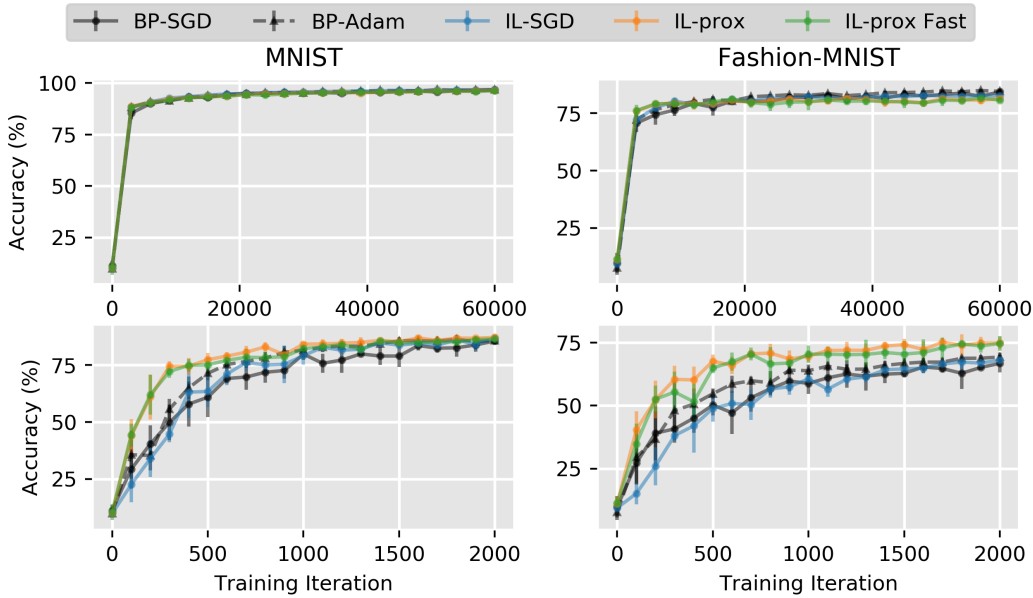

Figure 6: Training runs for MNIST and Fashion-MNIST. Networks were fully connected dimensions 784-2x500-10. Models were trained mini-batch size 1 for one epoch (60,000 iterations). **Top Left** Full training run for MNIST. **Bottom Left** A zoomed in look at the first 2000 training iterations of MNIST. IL-prox algorithm improve accuracy slightly faster than other algorithms in the first 500 iterations. **Top Left** Full training run for Fashion-MNIST. **Bottom Left** A zoomed in look at the first 2000 training iterations of Fashion-MNIST. IL-prox algorithms improve accuracy faster than other algorithms for first 2000 training iterations.

## F.4 Convolutional Networks

We also trained a small convolutional networks on CIFAR-10. Large mini-batches (size 64) were used and the Adam based algorithms were used for training (see 2), since IL algorithms worked best with Adam optimizers when large mini-batches were used. IL-prox Adam achieved essentially the same accuracy as BP-Adam, though IL-SGD did not achieve comparable accuracy.

| CIFAR-10 Test Accuracy (mean±std.) w/ Best Learning Rate | | |
| --- | --- | --- |
| BP-Adam | IL-Adam | IL-prox Adam |
| 66.850(±.591) | 54.802(±1.031) | 66.704(±1.725) |

Table 7: Here we train on CIFAR-10 with mini-batches of size 64 for 77,000 iteration ($\approx$ 100 epochs). Runs were averaged over five seeds and the best score is shown. There were three convolutional layers followed by one fully connected layer. We used a similar architecture as [3]. Convolutions were (5x5, 64, 2), (5x5, 128, 2), (3x3, 256, 2). Fully connected layer was 1024x10. ReLU activations were used at hidden layers, and softmax at the output layer.

## F.5 Data Constrained Tests

We test classification on F-MNIST and CIFAR-10 in the case where only 10, 100, or 500 data points are used from each category in the data set. Models are trained with mini-batch size 100. Grid search is used to find best learning rate. All models are trained to convergence and best accuracies, averaged across 5 seeds, are shown table 8. Training runs can be seen in figure 4 and best accuracies.

| Data Constrained Test | | | | | | |
|---|---|---|---|---|---|---|
| | F-MNIST | | | CIFAR-10 | | |
| Model | n=10 | n=100 | n=500 | n=10 | n=100 | n=500 |
| BP-SGD | 70.72($\pm$.44) | 79.41($\pm$19) | 84.00($\pm$.10) | 23.58($\pm$.58) | 33.18($\pm$15) | 44.54($\pm$.08) |
| IL-SGD | 72.94($\pm$.19) | 79.69($\pm$.11) | 83.63($\pm$.24) | 24.67($\pm$.65) | 34.63($\pm$.18) | 42.80($\pm$24) |
| IL-prox | 71.64($\pm$.19) | 78.90($\pm$.24) | 82.12($\pm$.16) | 24.00($\pm$.56) | 34.17($\pm$.20) | 39.80($\pm$.45) |
| BP-Adam | 70.99($\pm$.16) | 78.60($\pm$.19) | 84.09($\pm$.11) | 23.93($\pm$.24) | 34.11($\pm$.18) | 44.13($\pm$.25) |
| IL-Adam | 71.20($\pm$.29) | 80.52($\pm$.16) | 84.32($\pm$.11) | 24.07($\pm$.13) | 34.91($\pm$.23) | 44.41($\pm$.13) |
| IL-prox Adam | 70.96($\pm$.47) | 80.27($\pm$.083) | 84.16($\pm$.10) | 24.17($\pm$.29) | 34.58($\pm$.22) | 44.15($\pm$.17) |

Table 8: Top test accuracies in constrained memory scenario. Networks were trained on subsets of F-MNIST and CIFAR-10 data sets, i.e., where only 10, 100, or 500 data-points were used from each category. Mini-batch size 100 was used. 5 seeds were trained to convergence for each model and the best score from each seed were averaged.

### F.6 Output Layer Analysis

Theorem D.1 states that IL-GN, the closed form approximation of IL, updates parameters of linear MLPs in a way that changes FF output layer values ($h_N^{(b+1)} - h_N^{(b)}$) in the direction of the minimum-norm path of loss minimization for any positive learning rate. Theorem D.2 states that BP-GN updates parameters in a way that yields change in FF output layer values ($h_N^{(b+1)} - h_N^{(b)}$) in the direction of the minimum-norm path of loss minimization, but only for a finite range of positive learning rates. Here we test how well this result holds in practice in a network trained with IL-SGD. In particular, we test how closely change in FF output layer values $\Delta h_N = h_N^{(b+1)} - h_N^{(b)}$ align with the minimum norm path to the target $\Delta h_N^{min} = y - h_N^{(b)}$ using cosine similarity. For small linear networks trained on a regression task with, IL-SGD does indeed produce changes in the output (see 7 left) very close in angle to the minimum norm path. This supports the claim that IL-GN is a useful approximation of G-IL. MLPs trained with BP-SGD produce changes in the output layer that are significantly less close to the minimum norm path. This suggests IL-SGD takes a shorter, more direct path toward local minimum than that of steepest descent. Similar results were found when ReLU activations were applied at hidden layers (see 7 middle). Finally, we directly test theorem D.1 and theorem D.2 by exactly implementing IL-GN and BP-GN in a toy linear network. We measure the cosine similarity between the change in output FF values and the minimum norm change, as with the previous tests, and we do this over a range of learning rates. As predicted by theorem D.1 and D.2, IL-GN and BP-GN affect output layer values along minimum norm path toward the target. However, as predicted by theorem D.1 and D.2, we find BP-GN is significantly less stable than IL-GN (see 7 right).

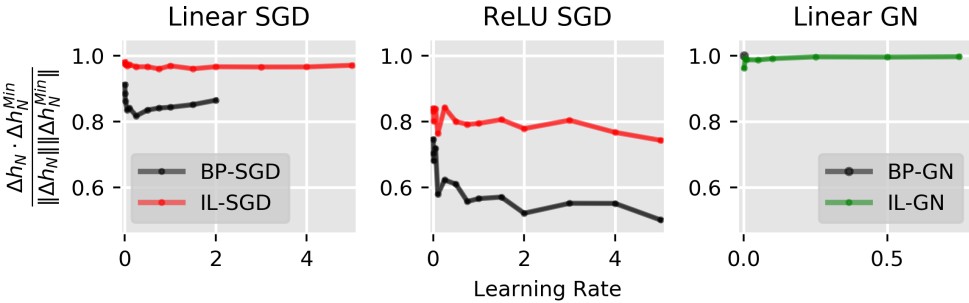

Figure 7: (Left) Cosine similarity was measured between $\Delta h_N$ and $\Delta h_N^{Min}$ each training iteration across a range of learning rates in linear networks trained on a regression task. $95\%$ confidence intervals are shown but are too small to be visible. (Middle) The same test was performed on non-linear networks that use ReLU at hidden layers. (Right) We run the same test for an actual IL-GN algorithm and BP-GN. As predicted by theorem D.1 and D.2 the IL-GN and BP-GN both push the output layer down its minimum-norm path, but BP-GN produces a minimum-norm $\Delta h_N$ only for a tiny range of learning rates (see black dot upper left), while IL-GN is stable over a much larger range.

Propositions E.1 and E.2 show that under certain assumptions IL-GN weight updates should have a larger effect on $h_N$ when applied together than when applied alone, and the opposite is generally true of BP-GN. We test how well this result holds in practice in IL-SGD and BP-SGD networks trained on a regression tasks. We compute the effect each individual $\Delta W_n^{(b)}$ has on the output layer FF values when applied in conjunction with other weights: $\|h_N^{(b+1)} - h_N^{(b+1/2,\neg n)}\|$, where $h_N^{(b+1)}$ are the FF output values after updates are applied to all weights, and $h_N^{(b+1/2,\neg n)}$ are the output layer values and when all updates except $\Delta W_n$ are applied. We then compute $\|h_N^{(b+1/2,n)} - h_N^{(b)}\|$, where $h_N^b$ are the FF output values before any updates are applied at the current training iteration, and $h_N^{(b+1/2,n)}$ are the output layer values after only $\Delta W_n$ is applied.

When $\|h_N^{b+1} - h_N^{(b+1/2,\neg n)}\|^2 \geq \|h_N^{(b+1/2,n)} - h_N^{(b)}\|$, we know $\Delta W_n$ had a larger affect on the output layer when applied in conjunction with other weights than when applied alone. When $\Delta W_n$ meets this condition we count it as compatible with the other weight updates. We then compute a compatibility score which stores a count of all compatible weight updates and uses it to compute the proportion of updates that were compatible.

$$CompatibilityScore = \frac{\#\text{Compatible } \Delta W_n}{\text{Total } \#\Delta W_n} \qquad (71)$$

We find that less than $50\%$ of weight updates in IL-SGD and BP-SGD networks are compatible though for smaller learning rates a greater proportion of IL updates are compatible than BP updates (see figure 8). This supports the claim that IL updates interfere with one another less than do BP updates, both in their ability to minimize loss and in their ability to change the FF output layer values.

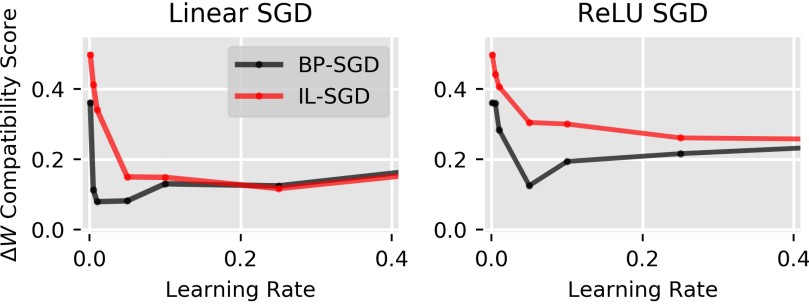

Figure 8: (Left) In linear networks with small learning rates ($\alpha_n < .1$) IL weight updates have significantly larger compatibility scores (according to 2-sample T-test, $95\%$ confidence intervals shown but are too small to see) than BP-SGD updates. (Right) The same trend exists in non-linear networks with ReLU at hidden layers for $\alpha_n < .4$.

### F.7 Weight Norm Analysis

Here we report the mean and standard deviation of the parameter updates $\|\Delta\theta\|$ from the training runs reported in 6. We can see that generally IL updates are on average smaller than BP-SGD updates and have smaller standard deviations. IL algorithms tended to improve accuracy faster than BP-SGD. This suggests that IL did not improve accuracy faster because the overall changes in parameters were larger. One possibility is that IL improved accuracy faster because it updated parameters along a more direct path toward local minima. This suggestion is consistent with the findings above, that IL updates push FF output layer activities along a more direct path toward global target activities $y$.

| $\|\Delta\theta\|$ (mean$\pm$std.) for Supervised Classification Tasks | | | |
|---|---|---|---|
| Model | MNIST | Fashion-MNIST | CIFAR-10 |
| BP-SGD | .0871($\pm$.0194) | .0680($\pm$.087) | .0807($\pm$.0284) |
| BP-Adam | .0109($\pm$.0082) | .0060($\pm$.0019) | .0090($\pm$.0014) |
| IL-SGD | .0074($\pm$.0165) | .0092($\pm$.0124) | .0036($\pm$.0018) |
| IL-prox | .0059($\pm$.0132) | .0361($\pm$.012) | .0149($\pm$.004) |
| IL-prox Fast | .0075($\pm$.0167) | .1215($\pm$.1102) | .0203($\pm$.0161) |

Table 9: Average weight update norms for classification tasks.

## G  Methods

In this section, we provide a more detailed description of the algorithms used in our simulation, as well as a more detailed description of the experiments, so that they may be replicated in the future. We begin by describing the algorithms in more detail then discuss experiment details.[3]

### G.1  Algorithms

**BP-SGD** This algorithm performed explicit SGD over weights computed by BP. Pytorch auto-differentiation was used to compute the gradients and update the weights. **BP-Adam** This algorithm computes (explicit) gradients using BP, which are then used by Adam optimizers to update parameters. We used the standard/default hyperparameters settings in the pytorch Adam optimizer: $\beta_1 = .9$, $\beta_2 = .999$, $\epsilon = 1 * 10^{-8}$.

**IL-SGD** This algorithm uses SGD during the inference phase and updates weights with a simple gradient/LMS step. The inference phase runs for 25 time/gradient steps over neuron activities. Consistent with previous work (e.g. [43, 2], we fully clamp the output layer, i.e., $\hat{h}_N = y$.

---

**Algorithm 3:** IL-SGD

**begin**
    // Feedforward Pass
    $\hat{h}_0 \leftarrow x^{(b)}$
    **for** $n = 0$ **to** $N - 1$ **do**
        |   $p_{n+1}, \hat{h}_{n+1} \leftarrow W_n f(h_n)$
    **end**
    $\hat{h}_N \leftarrow y^{(b)}$
    // Inference Phase (Compute Local Targets)
    $\hat{h}_n \leftarrow y^{(b)}$ **for** $i = 0$ **to** 25 **do**
        **for** $n = 1$ **to** $N$ **do**
            |   $\hat{h}_n \leftarrow \hat{h}_n - \beta \frac{\partial F}{\partial \hat{h}_n}$
            |   $p_{n+1} \leftarrow f(W_n \hat{h}_n)$
        **end**
    **end**
    // Update Weight Matrices
    Eqn. 4
**end**

---

Note that the predictions are computed $f(W_n \hat{h}_n)$, which is similar to previous implementations.

**IL-Adam** computes approximate implicit gradients of the weights using the IL-SGD algorithm (algorithm 3). These gradients are then used by Adam optimizers which actually perform the update. An Adam optimizer is assigned to each weight matrix. We used the standard/default hyper-parameters settings in the pytorch Adam optimizer: $\beta_1 = .9$, $\beta_2 = .999$, $\epsilon = 1 * 10^{-8}$.

---

[3]Code for simulations described below can be found at https://github.com/nalonso2/ILTheory

**IL-prox** This algorithm is our novel variant of IL that is made to more closely match the algorithm described in definition 4.1. Theorem 4.1 shows that minimizing the proximal loss w.r.t. hidden layer activities and output layer activities is equivalent to minimizing $F$ under certain settings of the $\gamma$ terms 4.1. The most important difference between these $\gamma$ settings and the ones of IL-SGD is the way the learning rate is used. In particular, IL-SGD scales weight updates using the learning rate, whereas IL-prox scales weight updates by the norm of presynaptic activities using the NLMS rule (equation 16) instead of learning rate. Then, following theorem 4.1, IL-prox uses the learning rate $\alpha$ only to determine the degree to which $h_N$ is pushed toward global target $y$ versus output layer prediction $p_N$. In particular, following theorem 4.1, $\gamma_N = \frac{\alpha_{N-1}}{\alpha} = \frac{1}{\|\hat{h}_{N-1}\|^2\alpha}$, then the gradient update at the output layer is

$$\Delta\hat{h}_N = \frac{\partial L}{\partial \hat{h}_N} - \frac{1}{\|\hat{h}_{N-1}\|^2\alpha}e_N, \tag{72}$$

where $e_N = \hat{h}_N - p_N$, $p_N = \sigma(W_{N-1}f(\hat{h}_{N-1}))$, and $\sigma$ is the softmax in classification tasks and the sigmoid in the autoencoder task. In practice, we use a cross-entropy loss to update weights but find using gradients of the cross entropy loss to update output layer activities leads to some instability. Instead, we use the gradient of the MSE to update output layer activities. In particular,

$$\Delta\hat{h}_N = -(y - \hat{h}_N) - \frac{1}{\|\hat{h}_{N-1}\|^2\alpha}e_N, \tag{73}$$

and we solve for $\hat{h}_N$ to get

$$\hat{h}_N = \frac{\|\hat{h}_{N-1}\|^2\alpha}{1 + \|\hat{h}_{N-1}\|^2\alpha}y + \frac{1}{1 + \|\hat{h}_{N-1}\|^2\alpha}p_N. \tag{74}$$

Thus, at each iteration of the inference phase, the output layer activity $\hat{h}_N$ is updated to a weighted average between global target $y$ and prediction $p_N$. We see that as $\alpha \to 0$ then $\hat{h}_N \to p_N$ and as $\alpha \to \infty$ then $\hat{h}_N \to y$. Interestingly, we see the inverse relation between $\hat{h}_N$ and the normalized learning rate. The normalized learning rate for $W_{N-1}$ is $\|\hat{h}_{N-1}\|^{-2}$. We see that as $\|\hat{h}_{N-1}\|^{-2} \to 0$ then $\hat{h}_N \to y$. This means that for very small normalized learning rates the output layer activity will be pushed toward the global target $y$, which will lead to larger output layer errors $e_N$. As $\|\hat{h}_{N-1}\|^{-2} \to \infty$ then $\hat{h}_N \to p_N$. This means that for very large normalized learning rates the output layer activity will be pushed increasingly fast toward the output layer prediction $p_N$, which will lead to smaller output layer errors. This illustrates how these activity updates manage weight update magnitude, as the magnitude of weight updates under the NLMS rule depend on the magnitude of post synaptic errors $e_N$ and the magnitude of the normalized learning rate. If the learning rate is small, $e_N$ can be larger and the weight update will still be small. If the learning rate is large, the activities are updated to keep $e_N$ small and thus the weight update small.

We find including the $\|\hat{h}_{N-1}\|^2$ term in the output layer activity update is important for the stability and performance of IL-prox, but the weighting terms of theorem 4.1 are not particularly important at hidden layers for performance or stability of IL-prox. To reduce computation, we just use preset hyper-parameter $\gamma$ values at hidden layers. IL-prox uses the NLMS rule (equation **??**) to update weights. In practice, we find it sometimes improves performance to add a small constant $\epsilon$ to the NLMS rule such that it becomes

$$\text{argmin}_{W_n} F = W_n + \Delta W_n = W_n + \frac{1}{\|f(\hat{h}_n)\|^2 + \epsilon}e_{n+1}f(\hat{h}_n)^T. \tag{75}$$

The $\epsilon$ provides a smooth upper bound on the magnitude of the normalized learning rate and prevents division by zero errors. (When using biases, however, division by zero is generally not an issue, since one must treat the pre-synaptic vector $\hat{h}_n$ as having a 1 concatenated to the end of it which for most activation functions guarantees $\|f(\hat{h}_n)\|^{-2} > 0$.)

In the case where mini-batches are used we first average the gradient across the batch, then multiply this averaged gradient by the average of the normalized learning rates across the mini-batch.

---

**Algorithm 4:** IL-prox

---

**begin**

    // Feedforward Pass

    $\hat{h}_0 \leftarrow x^{(b)}$

    **for** $n = 0$ **to** $N - 1$ **do**

        $p_{n+1}, \hat{h}_{n+1} \leftarrow W_n f(h_n)$

    **end**

    $\hat{h}_N \leftarrow$ Eqn. 74

    // Inference phase (Compute Local Targets)

    **for** $i = 0$ **to** $25$ **do**

        **for** $n = 1$ **to** $N$ **do**

            $\hat{h}_n \leftarrow \hat{h}_n - \beta \frac{\partial F}{\partial \hat{h}_n}$

            $p_{n+1} \leftarrow W_n f(\hat{h}_n)$

        **end**

        $\hat{h}_N \leftarrow$ Eqn. 74

    **end**

    // Update Weight Matrices

    Eqn. 75

**end**

---

**IL-prox Fast** is the same as algorithm 4 except it only performs 12 updates over activities. **IL-prox Adam** uses IL-prox to compute (approximate implicit) gradients (i.e. uses the same inference phase as IL-prox then equation 75 to compute weight gradients), then inputs these gradients to an Adam optimizers, which updates the weights.

**BP-prox** was the algorithm we used as a control to compare against IL-prox in the stability tests. BP-prox softly clamps the output layer similarly to IL-prox and uses the NLMS rule similarly to IL-prox. The output layer activities are computed according to

$$\hat{h}_N = \frac{\|\hat{h}_{N-1}\|^2 \alpha}{1 + \|\hat{h}_{N-1}\|^2 \alpha} y + \frac{1}{1 + \|\hat{h}_{N-1}\|^2 \alpha} h_N. \tag{76}$$

Notice, unlike IL-prox, the FF activity $h_N$ is used rather than $p_N$ since G-BP does not compute local predictions. However, IL-prox initializes $p_N$ to $h_N$ so these values are the same upon initialization. Here $h_N = \sigma(W_{N-1} f(h_{N-1}))$ where $\sigma$ is the softmax. The weights are updated as follows:

$$\Delta W_n = \frac{1}{\|f(h_n)\|^2 + \epsilon} e_{n+1} f(h_n)^T, \tag{77}$$

where $\epsilon$ is a small scalar used to prevent division by zero and place upper bound on the normalized learning rate. (As we note below, in the stability tests the same small $\epsilon$ is used for both IL-prox and BP-prox for a fair comparison). Notice this is the same as the NLMS rule used by IL-prox (equation 75), except, in keeping with the G-BP framework, the pre-synaptic FF activities $h_n$ activities are used in the learning rate and gradient com-

putation rather than target activities $\hat{h}_n$. A summary of the algorithm is presented below.

---

**Algorithm 5:** BP-prox

---

**begin**
 // Feedforward Pass
 $h_0 \leftarrow x^{(b)}$
 **for** $n = 0$ **to** $N - 1$ **do**
  |  $h_{n+1} \leftarrow W_n f(h_n)$
 **end**
 $\hat{h}_N \leftarrow$ Eqn. 76
 // Compute Local Targets
 **for** $n = 1$ **to** $N$ **do**
  |  $\hat{h}_n = h_n - \frac{\partial L(y, \hat{h}_N)}{\partial h_n}$
 **end**
 // Update Weight Matrices
 Eqn. 77
**end**

---

## G.2   Experiment Details

Pytorch data-loaders were used to download and perform transformations on MNIST, Fashion-MNIST, and CIFAR-10 datasets. All data was transformed to pytorch tensors, but no other alterations were applied to the data (e.g., there was no normalization). All pixel values were in the range between 0 and 1. Pytorch auto-gradient library was used to compute gradients for BP models, and the local gradients for IL-SGD models. Gradients used to update weights and activities in IL-prox models were computed manually. Code is publicly available at ???.

**Supervised Tasks** For all classification tasks, ReLU activations were used at hidden layers and softmax at the output layer. For MNIST tasks, fully connected networks size 784-2x256-10 were used. For our main results with CIFAR-10 classification (table 2), we used fully connected networks size 3072-3x1024-10. For the convolutional classification we used a similar architecture as [3], who compared a wide range of local,biologically plausible learning algorithms. Convolutions were (5x5, 64, 2), (5x5, 128, 2), (3x3, 256, 2). Fully connected layer was 1024x10. (Note, these are the same dimensions as [3], though they used locally connected layers rather than convolutions.)

A grid search was used to find the learning rates for CIFAR-10 and MNIST classification tasks shown in tables 2, 6, and 7. We first hand tuned the learning rates to find a range over which the algorithms performed the best, then chose 5-10 learning rates in the range and trained 2-5 seeds at each learning learning rate. We averaged the test accuracy achieved at the end of training over the seeds, then chose the learning rate with the highest average test accuracy at the end of training. We then reran the training run with the best learning rate across 5 seeds to get the data shown in tables 2, 6, and 7. Each seed was tested every 50 training iterations. Runs were averaged over the 5 seeds. The best test accuracy achieved from the averaged training run, is shown in the tables.

Learning rates used for each algorithm are shown below. The other main hyper-parameters are the $\gamma$ terms in the energy equation (equation 4), the gradient step size used to update activities in the IL models, and the $\epsilon$ values in the IL-prox update (equation 75). The $\gamma$ terms and the step sizes over activity updates in IL models, each scale the gradients of F. To simplify, we found it effective to just set $\gamma^{decay} = 0$, then use the same two hyper-parameter terms at each hidden layer to differentially weight the two error gradient terms: $\gamma^{bot}$ and $\gamma^{top}$. First, $\gamma^{bot}$ is multiplied by the 'bottom up' error gradient in equation 24, which comes to $\gamma^{bot} f'(\hat{h}_n) W_n^T e_{n+1}$. Then we $\gamma^{top}$ is multiplied by the 'top-down' error gradient in equation 24, which comes to $\gamma^{top} e_n$. These same two hyper-parameters are used at each hidden layer of IL models and can be seen as incorporating both the actual gamma and step size hyper-parameters into a single hyper-parameter. For IL-SGD and IL-Adam, $\gamma^{bot} = .02$ and $\gamma^{top} = .015$. For IL-prox and IL-prox Adam, $\gamma^{bot} = .015$ and $\gamma^{top} = .015$. For IL-prox Fast, $\gamma^{bot} = .015$ and $\gamma^{top} = 0$. Finally, $\epsilon = .25$ for IL-prox and IL-prox fast models.

**Self-Supervised Tasks** The same grid search procedure was used for the self-supervised task as for the supervised tasks to find learning rates for each algorithm. ReLUs were used at hidden layers and

| Learning Rates - Supervised Task - Fully Connected Networks | | | | | |
|---|---|---|---|---|---|
| Model | MNIST | Fashion-MNIST | CIFAR-10 | CIFAR-10 (mb=64) | CIFAR-10 Conv. (mb=64) |
| BP-SGD | .015 | .01 | .015 | .01 | - |
| BP-Adam | .0001 | .00005 | .000025 | .00005 | .00002 |
| IL-SGD | .05 | .03 | .01 | .01 | - |
| IL-prox | 5 | 2.5 | 100 | 100 | - |
| IL-prox Fast | 3 | 2.5 | 100 | 100 | - |
| IL-Adam | – | – | .00001 | .000005 | .000009 |
| IL-prox Adam | – | – | 100 | 100 | .000008 |

Table 10: Learning rates for algorithms trained on supervised learning tasks with fully connected networks, with various data sets and mini-batch sizes (mb).

sigmoid at the output layer. The autoencoder used fully connected layers dimensions 3072-1024-500-100-500-1025-3072. Binary cross entropy (BCE) was used at the output layer. The BCE data shown in figure 3 was averaged over pixels and averaged over the test set. The best test score from each of these averaged training runs is shown in the table. Figure 3 shows a run with mini-batch size 1, which were trained with 50,000 data point (1 epoch). Learning rates are shown in the table below. The $\gamma$ settings are the same as those from the previous section. For the NLMS rule in IL-prox we used an $\epsilon = 20$ and generally found it useful to have a larger $\epsilon$ (and thus a smaller upper bound on the normalized learning rate) than in the supervised task.

| Learning Rates - Self-Supervised Task | | | | |
|---|---|---|---|---|
| BP-SGD | BP-Adam | IL-SGD | IL-prox | IL-prox fast |
| .0001 | .00005 | .04 | 10 | 10 |

Table 11: Learning rates for self-supervised task

**Stability Test** Here we train models over learning rates .01, .1, 1, 2.5, 10, 100. We use the BP-SGD, IL-SGD, BP-prox, and IL-prox algorithms to train 5 seeds of networks sized 784-2x256-10 for MNIST and 3072-3x1024-10 for CIFAR-10, and use ReLU at hidden layers with softmax at the output layer. We use mini-batch size 1 and train for 50,000 iterations.

Importantly, the accuracy shown in table 4 and 5 are the test accuracy, averaged over the 5 seeds, at the end of training. We use the test accuracy at the end of training (rather than the best test accuracy achieved during training). The purpose of this was to test which algorithms not only achieved above chance accuracy, but also did so while converging in a stable manner. Also important, the $\epsilon$ values used in the IL-prox and BP-prox updates were set to zero to ensure any stability these algorithms showed in this test was not due simply to the $\epsilon$ down scaling the weight updates.

**Output Layer Analysis** For the output layer analyses presented in figures 7, 8 we trained small toy models on a regression task. Models were FF networks with layer sizes, 10-5-5-5-5. Input data was generated by sampling vectors from a standard Gaussian. A teacher network of the same size with Tanh activations at hidden layers, was used to generate associated output target vectors associated with each input vector.

Linear toy networks and toy networks with ReLUs at hidden layers were trained with BP-SGD and IL-SGD across learning rates (.001, .005, .01, .05, .1, .25, .5, .75, 1, 1.5, 2, 3, 4, 5). Each algorithm trained 50 seeds at each learning rate for 200 training iterations. Each training iteration the compatibility score (equation 71) was computed. Additionally, each training iteration $b$, we computed the minimum norm path between the output layer activities and the output target $y^{(b)}$ as $y^{(b)} - h_N^{(b)}$. We then compared this value to the actual change in the output layer values $h_N^{(b+1)} - h_N^{(b)}$ after the update each training iteration. We then computed and stored the cosine similarity between the two. This data was then averaged over each iteration and seed for each learning rate. The result was the average compatibility score and minimum norm change similarity under each learning rate.