# OpenReview forum: "A Theoretical Framework for Inference Learning"
_NeurIPS.cc/2022/Conference — NeurIPS 2022 Accept_

### Official Review · Reviewer_an3X · 2022-07-07

**Rating:** 7
**Confidence:** 2
**Soundness:** 3 good
**Presentation:** 3 good
**Contribution:** 4 excellent

**Summary:**

The paper gives a theoretical framework and justification for the biologically plausible "inference learning" neural-network training algorithm, a generalization of predictive coding. The paper shows that in the low-N regime, IL closely approximates implicit stochastic gradient descent: it steps in the gradient direction while also staying close to the current weights. Based on this theorem, the paper then improves IL to more closely approximate implicit SGD, which improves the stability and convergence of IL across learning rates and tasks.

**Questions:**

Have the authors tested against implicit SGD with a backprop-based gradient estimator?

**Limitations:**

I can't quite see where this paper fits in.  Bioplausible training of artificial neural networks in the small-batch (eg: N=1) regime remains, to my knowledge, not a very common task.  If the idea is that the brain might do something like IL, can the authors point to any experiments specifically picking out IL as distinct from predictive coding in the brain?  If the idea is to train ANNs for tasks, why not just use BP and large batch-sizes?  Have the authors run any experiments in memory-constrained settings showing that IL does better there?

The authors have addressed these questions to my satisfaction in their response.

**Strengths And Weaknesses:**

Strengths:

* Strong bioplausibility motivation
* Beats backpropagation in the small-batch setting

Weaknesses:
* Beaten by backprop in the midsize-batch (N=64) setting

---

> ### Author Response · Authors · 2022-08-02
> **Response to an3X's Questions and Concerns**
>
> We thank reviewer an3X for their questions and comments. The reviewer an3X first asked ‘Have the authors tested against implicit SGD with a backprop-based gradient estimator?’ We are not entirely sure what the reviewer means. Clarification would be appreciated.
>
> The reviewer states in the limitation section that they ‘cannot see where this paper fits in’ given that biologically plausible training in scenarios with mini-batch size one is not a common machine learning task. Relatedly, the author asks why anyone would want to use IL to train neural networks on tasks, if we can use mini-batches to speed up training and if BP works better than IL with mini-batches? We agree these are important questions/concerns. In the revised draft we make clearer why we think these results are interesting both for the neuroscience and machine learning community. Here’s a more detailed explanation:
>
> 1) Within the computational neuroscience community, we believe our paper makes a clear contribution, as it provides a novel analysis of an algorithm used to train a popular neural model, and the theoretical analysis suggests a novel hypothesis about what optimization strategy is used by the brain, i.e., implicit stochastic gradient descent. Additionally, through theoretical and experimental results, it also characterizes how this optimization strategy behaves differently than backpropagation. These theoretical results focus on the case of mini-batch size one, but this is the situation the brain faces (the brain does not receive mini-batched inputs from the environment). According to the neurips 2022 official call for papers page, it is written that neuroscience submissions are invited.
>
> 2) For the machine learning community, biologically plausible, local learning algorithms like IL are of interest to neuromorphic computing research, as the standard implementation of BP is somewhat incompatible with energy efficient neuromorphic hardware. This has led to an interest in bio-plausible algorithms that use local learning rules and may be more compatible with neuromorphic constraints. Additionally, although networks can be pre-trained with mini-batches before being run on neuromorphic hardware, it is often said that online learning in neuromorphic hardware is desirable. This may not be a common task yet, but may be in the future. Finally, we do not intend the conclusion of our work to be that researchers should right now apply IL on standard (non-neuromorphic) machine learning tasks to achieve SOA results. Rather, we intend our work to motivate research into finding ways to improve IL’s performance and to test its performance on a variety of tasks and further compare against BP. Decades of work by many researchers have been put into improving BP, while only several years of work by a handful of researchers have gone into applying IL to machine learning tasks. We believe our research suggests that IL may be developed into a high performing algorithm that may be competitive with BP and possibly advantageous in some (though not all) scenarios (e.g., neuromorphic, lifelong learning) and is thus worth developing further. We hope our theoretical foundations provide a basis for such work. We added some of these comments to the revised version of the paper in the intro and conclusion.
>
> The author asks whether there is any evidence that the brain does IL (i.e., implements the IL learning algorithm) in particular, apart from evidence the brain just does predictive coding (i.e., implements the predictive coding neural architecture)? This is an important question from the neuroscience side. Interestingly, a paper was posted on bioarxiv recently by Yuhang Song et al. is called ‘Inferring Neural Activity Before Plasticity: A Foundation for Learning Beyond Backpropagation’. This paper uses neural evidence to support the idea that the brain first updates neural activities through an inference process, then updates synapses so that those updated/target neural activities are reproduced more easily the next time the same input is received, just like inference learning. More evidence is needed of course, but we believe this provides nice direct evidence in support of the hypothesis the brain does something like IL. Their paper does not make a formal connection between IL and implicit sgd like we do, however. Our work may provide a mathematical explanation of what the data is showing. We added this citation to our revised version.
>
> The reviewer asks if we have run any memory-constrained scenarios, to see if IL outperforms BP in this scenario. We had not run any memory-constrained scenarios originally. After receiving this question, we did test memory constrained scenarios and found IL did perform better than BP in highly constrained memory. These new results were added to the appendix and mentioned in results section. See response to reviewer nxGR for a summary of other edits and additions.

---

> > ### Comment · Reviewer_an3X · 2022-08-03
> > **Thank you authors for your clarifications**
> >
> > Where I asked if the authors have tested against implicit SGD with typical gradient estimators, I meant that there are proximal optimization algorithms that can exploit automatic differentiation to perform implicit-SGD updates, as well as certain statistical models in which implicit-SGD updates even have closed form (https://projecteuclid.org/journals/annals-of-statistics/volume-45/issue-4/Asymptotic-and-finite-sample-properties-of-estimators-based-on-stochastic/10.1214/16-AOS1506.full).  Have the authors IL updates against such implicit SGD updates?
> >
> > I take the authors' suggestions regarding neuromorphic hardware to heart, and in light of their responses will be raising my rating.

---

> > > ### Author Response · Authors · 2022-08-10
> > > **Follow Up to Reviewer an3x**
> > >
> > > We thank reviewer an3x for their clarification and are happy to hear they found our response compelling. As for the reviewer's question, we have not tested IL updates against implicit SGD updates that exploit automatic differentiation, but we think that would be a useful comparison for us to do in future work. We have found closed form solutions for implicit SGD for linear and generalized linear models (as in the paper cited by an3x) but are unaware of closed form solutions for deep networks. However, integrating a closed form update into the local learning rules of IL, especially in the mini-batch scenario, would be interesting to test in future research.

---

### Official Review · Reviewer_XkhF · 2022-07-11

**Rating:** 6
**Confidence:** 4
**Soundness:** 2 fair
**Presentation:** 2 fair
**Contribution:** 2 fair

**Summary:**

This paper proposes an alternative to gradient back propagation in the form of inference learning. This algorithm has the advantage of being more compatible with a biophysical implementation, and the paper aims to develop a theoretical framework for this learning. The expected results are to obtain convergence results for this algorithm and to show the performance compared to classical BP algorithms. The abstract makes a very strong assumption about the result by stating that this algorithm can be superior to the BP algorithm.


**Questions:**

I have several questions about this work.
Firstly, from equation (1) the cost function used to calculate an optimal prediction is always an L2 norm. How can you justify this, and have you verified this result experimentally by looking at the distribution of errors in learning?
Secondly, in generalized inference learning, you introduce a decay term in equation (3). Could you justify more accurately the evidence for this finding in the simulation data?
Third, could you substantiate a comparison of your algorithm with hierarchical predictive coding algorithms that include both a sparseness regularization or a feedback term?


**Limitations:**

Overall, the paper suffers from a major limitation between the conclusions that are put forward in the abstract and the theoretical but also simulation results that are presented in the paper. It would surely have been more correct to make less far-reaching assumptions but fully validated by the theoretical and experimental results.

**Strengths And Weaknesses:**

A main strength of the paper is to show the state of the art in machine learning algorithms, and in particular to show the comparison between stochastic gradient descent type models and inference learning models. The model is well presented, and the notations are relatively clear. In particular, the main limitation of the back propagation algorithm is well highlighted by showing how to implement a local target estimation algorithm.

A first weakness of the paper is that it relies on details that are supplied in the (40 pages) appendix and that relying on this does not allow an independent evaluation of the results in the paper. Secondly, even if the results seem encouraging, they are still clearly insufficient to show the generality of the algorithm. Indeed, the only benchmark used is that of CIFAR and the results are obtained with results superior to BP only in the very restrictive case of a batch size equal to one. This simulation result does not therefore justify the conclusion made in the summary about the generality of the results.

---

> ### Author Response · Authors · 2022-08-02
> **Response to Reviewer XkhF's Questions and Concerns**
>
> We thank reviewer XkhF for their comments and for citing their concerns.  XkhF’s main concern is that the generality of our main claims were not justified by the simulations we ran. First, the reviewer stated these results did not establish the generality of our claims, in part, because “the only benchmark used is that of Cifar-10”. However, we did train our networks on MNIST and F-MNIST (see appendix F.3) classification tasks. We apologize these results may have been overlooked since they were in the appendix. We found that IL discounted the loss quicker than BP early in training when mini-batch size one is used. Thus, we believe these results do generalize to other datasets. To further validate these results, we train another autoencoder on F-MNIST with mini-batch size 1 and found a significant speed up in training over BP-SGD. We show these results along with the classification F-MNIST results, in a new plot along with CIFAR-10 results in main experiments section. See our response to reviewer nxGR for a further summary of edits and additions.
>
> The reviewer also stated we did not establish that IL generally converged quicker than BP, in part, because this behavior was observed “only in the very restrictive case of a batch size equal to one”. However, in the abstract we only say “...IL achieves quicker convergence when trained with small mini-batches”. In the revision, we further clarify by switching ‘small mini-batch’ to ‘mini-batch size 1’ in the abstract and intro.
>
> It is also worth noting that the speed up in learning of IL over BP in small mini-batch scenarios is a secondary result. Our main result is the theoretical result that IL closely approximates implicit gradient descent. This was proven in the appendix. One assumption of the proof was indeed that mini-batch size one is used. However, we only stated in the abstract that “...IL closely approximates … implicit stochastic gradient descent” in order to take into account the fact that our main proofs are based on the assumption of mini-batch size 1, and thus the proofs only apply approximately to the case of mini-batch size >1. Thus, we do not believe we overstated the generality of our main theoretical result in the abstract either. In the revision, we further clarify by stating this approximation to implicit SGD holds closest in the case of mini-batch size 1.
>
> Question 1: The reviewer asked us to justify using the L-2 norm in equation 1, and asked whether we justified this experimentally by looking at the distribution of errors. Response: In the algorithm described in that section, local targets are computed as a global loss gradient step over neuron activities. Under this assumption when one computes the gradient of the L-2 norm of the difference between these targets and the activities w.r.t. weights, one gets back a weight update equivalent to BP. Prior work has proved this to be true (see, e.g., the appendix of the Bartinuv et al. reference in our paper). The goal of this section was to develop a target based algorithm equivalent to BP for easier comparison to IL.
>
> Question 2: The reviewer asked if we could justify the use of the decay term in our energy equation, and if we could point out how the simulation data justifies it (equation 3). Response: We included the decay term because the decay term is often used in predictive coding models (e.g., see the classic model of Rao and Ballard (1999)) and the decay term is also directly derived from our mathematical proofs (see proof for theorem 4.1 in appendix). Theorem 4.1 shows that in order to exactly equal implicit gradient descent, IL needs to perform some (usually small) decay over neuron activities during the inference phase. We would be interested in hearing clarification in what is meant by justifying the decay term with simulation data.
>
> Question 3: The reviewer asks us ‘’could you substantiate a comparison of your algorithm with hierarchical predictive coding algorithms that include both a sparseness regularization or a feedback term?’’. Response: As we use it, the term predictive coding refers to a kind of recurrent neural network architecture. Inference learning is the learning algorithm typically used to train predictive coding network architectures. Our generalized-IL is general with respect to the way neuron activities are updated to minimize energy, whereas standard models almost always use stochastic gradient descent. Although our theoretical results extend beyond these standard models, we do use SGD to update neuron activities in our simulations, which requires the use of both feedforward and feedback connections (see appendix section G). Our models do not use a sparseness constraint, but seeing if sparsity is compatible with the implicit gradient descent interpretations would be an interesting future avenue for research.

---

### Official Review · Reviewer_nxGR · 2022-07-11

**Rating:** 7
**Confidence:** 3
**Soundness:** 3 good
**Presentation:** 3 good
**Contribution:** 3 good

**Summary:**

This paper proposes a general theoretical framework for inference learning. To this end, the authors propose the Generalized Inference Learning (G-IL) algorithm. They are then able to show that G-IL closely approximates the implicit stochastic gradient descent (implicit SGD), which corresponds to the proximal update of the weight parameters and is distinct from the explicit SGD often used by backpropagation. In addition, the authors compare the performance of several algorithms based on backpropagation with those based on inference learning and establish that the latter converges more quickly on smaller mini-batches.

**Questions:**

While the authors claim that inference learning "has achieved equal performance to BP on supervised learning and auto-associative tasks", there is a clear discrepancy between the test accuracy of BP-SGD and IL-SGD with a mini-batch size of 64. While the authors mention this in their results, I was wondering where this non-negligible discrepancy comes from and what its implications for the comparison between inference learning and backpropagation would be.

**Limitations:**

The main limitation of the paper is that it mainly explores the case where a single data point is present in each mini-batch. The authors bring up this limitation in their work and provide a justification as to why it is not significant from a biological point of view. As such, I believe the authors adequately address the limitations of their work. However, I am not sure whether exploring the properties of larger mini-batch sizes in more detail could still yield some interesting results.

**Strengths And Weaknesses:**

To the best of my knowledge, this paper offers a novel perspective on inference learning by establishing the connection between G-IL and implicit SGD. The authors do a good job of motivating the similarities between inference learning and biological learning processes. In addition, they provide both theoretical justifications and experimental results for their claims.

As for the weaknesses, I believe that the authors could explain or provide some ideas for some of the experimental results they observe in more detail (such as IL-SGD performing worse than BP-SGD, but IL-Adam and BP-Adam performing more similarly on CIFAR-10).

---

> ### Author Response · Authors · 2022-08-02
> **Response to nxGR's question concerning discrepancy between results with and without Adam**
>
> We thank reviewer nxGR for their comments. The main concern of this reviewer was that they would like to see a more in depth discussion of why we saw some of the empirical results we did, in particular, ‘IL-SGD performing worse than BP-SGD, but IL-Adam and BP-Adam performing more similarly on CIFAR-10’ when mini-batch size 64 is used.
>
> We agree that this is an important question that could be addressed in more detail in the paper. We rewrote our discussion section so that it focuses more on this particular question (see second paragraph of discussion). We cannot provide a certain or complete answer since our theoretical results do not account for the parameter-wise adaptive learning rates and momentum that Adam uses, but we have a hypothesis that is consistent with our results.  In sum, our results, and prior work on implicit sgd in linear regression models (e.g., see Parikh & Boyd, 2014) show that implicit gradient descent pulls parameters along a more direct, shorter path toward local minima and uses updates that are smaller in magnitude than explicit sgd/BP. The shorter path probably explains why we often see a speed up in learning early in training. We also believe that IL’s ability to minimize the magnitude of weight updates acts as a sort of regularization that helps its performance in scenarios where data is noisy (e.g., mini-batch size one). However, in large mini-batch scenarios this regularization will provide less of an advantage over BP, and the short/more direct path IL takes toward local minima may mean IL pulls parameters into nearby shallow minima, instead of further away deeper minima. When Adam is used, however, the momentum and adaptive learning rates may aid IL by pushing it out of these shallow local minima into deeper local minima.
>
> Reference
> Parikh, N., & Boyd, S. (2014). Proximal algorithms. Foundations and trends® in Optimization, 1(3), 127-239.
>
>
> Below we summarize the edits made to the paper, and which review motivated the edit:
>
> -Trained autoencoders on F-MNIST with IL and BP algorithms and added new results, along with other F-MNIST results in appendix, to main results section with cifar-10 results in new plot (XkhF)
> -Edited wording in abstract and introduction to make more clear the speed up in training is observed only in mini-batch size 1 scenario and the theoretical results are based on this assumption (XkhF)
> -Performed memory constrained tasks on IL and BP networks, and showed results in appendix. We found IL performed better than BP on the highly memory constrained scenarios (an3X)
> -Added Yuhong et al. (2022) citation and several sentences to intro and conclusion on the evidence this paper provides in support of hypothesis the brain does IL (an3X).
> -Added several sentences to the introduction that further motivated and clarified the importance of our results for neuroscience and machine learning (an3X).
> -Added a few sentences to discussion, hypothesizing why IL-SGD and IL-prox did slightly worse than BP-SGD with large mini-batches, but IL-Adam performed similarly to BP-Adam with large mini-batches (nxGR).

---

> > ### Comment · Reviewer_nxGR · 2022-08-09
> > **Thanks for the response**
> >
> > I thank the authors for their response and acknowledge that they have positively addressed my comments.

---

### Meta-Review · Area_Chair_3p8J · 2022-08-29

**Recommendation:** Accept
**Confidence:** Certain

**Metareview:**

This paper presents an interesting connection between stochastic gradient descent by backpropagation and the "inference learning" algorithm for predictive coding. The key result is that inference learning approximates _implicit_ gradient descent, rather than explicit SGD as normally implemented. The implicit methods perform comparably to standard methods, and they may be of interest to computational neuroscientists interested in biologically plausible learning rules.

In addition to addressing the reviewers' concerns, I would encourage the authors to improve the exposition around Eqs. 1 and 2. The stated equalities require a few lines of calculus to derive, and you could spare the reader the trouble.

**Award:**

No

---

### Decision · Program_Chairs · 2022-09-14

Accept